



# Very high stratospheric influence observed in the free troposphere over the Northern Alps – just a local phenomenon?

Thomas Trickl[1], Hannes Vogelmann[1], Ludwig Ries[2], and Michael Sprenger[3]

[1]Karlsruher Institut für Technologie, Institut für Meteorologie und Klimaforschung, IMK-IFU, Kreuzeckbahnstr. 19, 82467 Garmisch-Partenkirchen, Germany
[2]Umweltbundesamt II 4.5, Plattform Zugspitze, GAW-Globalobservatorium Zugspitze-Hohenpeißenberg, Schnee-fernerhaus, 82475 Zugspitze, Germany
[3]Eidgenössische Technische Hochschule (ETH) Zürich, Institut für Atmosphäre und Klima, Universitätstraße 16, 8092 Zürich, Switzerland

*Correspondence to:* Dr. Thomas Trickl, thomas.trickl@kit.edu, Tel. +49-8821-183-209, Fax +49-8821-73573

**Abstract.** The atmospheric composition is strongly influenced by a change in atmospheric dynamics, which is potentially related to climate change. A prominent example is the doubling of the stratospheric ozone component at the summit station Zugspitze (2962 m a.s.l., Garmisch-Partenkirchen, Germany) between the mid-seventies and 2005, roughly from 11 ppb to 23 ppb (43 %). Systematic efforts for identifying and quantifying this influence have been made since the late 1990s. Meanwhile, routine lidar measurements of ozone and water vapour carried out at Garmisch-Partenkirchen (German Alps) since 2007, combined with in-situ and radiosonde data and trajectory calculations, have revealed that stratospheric intrusion layers are present on 84 % of the yearly measurement days. At Alpine summit stations the frequency of intrusions exhibits a seasonal cycle with a pronounced summer minimum that is reproduced by the lidar measurements. The summer minimum disappears if one looks at the free troposphere as a whole. The mid- and upper-tropospheric intrusion layers seem to be dominated by very long descent on up to hemispheric scale in an altitude range starting at about 4.5 km a.s.l. Without interfering air flows, these layers remain very dry, typically with RH ≤ 5 % at the centre of the intrusion. Pronounced ozone maxima observed above Garmisch-Partenkirchen have been mostly related to a stratospheric origin rather than to long-range transport from remote boundary layers. Our findings and results for other latitudes seem to support the idea of a rather high contribution of ozone import from the stratosphere to tropospheric ozone.

*Key words:* Ozone, water vapour, aerosol, stratosphere-to-troposphere transport, transport modelling, lidar, LAGRANTO

## 1 Introduction

For many years the pronounced rise of tropospheric ozone due to the growing anthropogenic air pollution has been the subject of intensive research. The background level of ozone has reached 50 ppb and more at some sites in the northern mid-latitudes (e.g., Parrish et al., 2012). However, megacity ozone may reach even several hundred ppb (e.g., Parrish et al., 2011; 2016) that may ultimately contribute to the tropospheric ozone background.

On the other hand, the most important natural source of tropospheric ozone, i.e., the import from the stratosphere, has been frequently related to the Montsouris value of just about 10 ppb estimated for the late 19th century (Volz and Kley, 1987). Layers of stratospheric air can be identified directly based on criteria like eleva-



ted ozone and low humidity. This direct import in deep stratospheric air intrusions has long resulted in estimates of the stratospheric influence on the tropospheric ozone level of about 10 % and less (e.g., Elbern et al., 1997; Beekmann et al., 1997; Stohl et al., 2000). This would suggest a rather small relative importance of stratosphere-to-troposphere transport (STT), with some uncertainty originating from the "indirect" stratospheric component

that cannot be detected due to complete mixing of the stratospheric intrusions into the troposphere

Quantifying STT has been attempted for more than half a century. Whereas early studies aimed at identifying the STT mechanisms (e.g., Danielsen, 1968) more recent work has also estimated the STT budget by extrapolations of observational data (e.g,, Danielsen and Mohnen, 1977; Viezee et al., 1983; Beekmann et al., 1997) or by diagnosing the cross-tropopause transport by global and regional weather and climate models (e.g., Roelofs and

Lelieveld, 1997; Kentarchos and Roelofs, 2003; Stevenson et al., 2006; Wild, 2007; Young et al., 2013). Model-based approaches have frequently concentrated on the overall exchange rate (for STE) rather on than that for STT. In the most recent multi-model comparison (Young et al., 2013) the models agree within about ±20 % (standard deviation) around an average net STE rate of 477 Tg a$^{-1}$. This value, obtained as a difference of the steady-state photochemical production rate and the loss rates, is just about 10 % of the production rate. The

ozone mixing ratio due to STT was, e.g., obtained from semi-Lagrangian approaches such as by Roelofs and Lelieveld (1997) or Collins et al. (2003). Roelofs and Lelieveld (1997) found a high fraction due to STT of 40 % in tropospheric ozone. This value looks rather high given the coarse resolution of the underlying chemistry-transport model of (e.g., horizontally) 3.75º × 3.75º that is insufficient for resolving thin stratospheric layers in the troposphere (see Roelofs et al., 2003; Rastigejev et al., 2010; Eastman and Jacobs, 2017). The stratospheric contribu-

tions of Collins et al. (2003) are lower (roughly 30 % above mid-latitude sites, the average mixing ratio typically rising from 40 to 60 ppb from the lower to the upper troposphere) and vary from site to site. In a study with higher spatial resolution Jaeglé et al. (2017) examined STT in dry intrusions associated with extratropical cyclones. They found that, on average, 15 % of the ozone mass in a dry intrusion is irreversibly mixed into the troposphere.

In principle, a quantification of STT should involve observations. However, determining the STT flux from

observations is a highly demanding task. Assessments from observations is easier for STT than for TST (troposphere-to-stratosphere transport) since both ozone and water vapour are suitable complementary tracers. The results strongly depend on the criteria selected for data filtering based on tracers such as $O_3$, $H_2O$ or $^7Be$ (e.g., Stohl et al., 2000). In addition, mixing of the dry descending layers with tropospheric air must be taken into account, which was, however, recently found to be much less severe than previously thought (Trickl et al., 2014; 2016).

Gradually, STT has turned out to be potentially much more important than concluded from the early assessments. For instance, a correlation study of $O_3$ and $H_2O$ based on vertical profiles derived from aircraft ascents and descents in the vicinity of airports within the MOZAIC (Measurement of Ozone, Water Vapour by Airbus In-Service Aircraft, Marenco et al., 1998) project revealed a rich ("ubiquitous") layer structure in the free troposphere mostly of the northern hemisphere (Newell et al., 1998; Thouret et al., 2000). The layers were

identified by positive or negative departures from mean background. The layers with excess ozone and reduced water vapour clearly dominate with about 50 % of the cases in all regions. Although the threshold for relative humidity drop applied (5 %) does by far not match the expectations for STT (Trickl et al., 2014; 2015) this high fraction is a hint towards significant downward transport. The occurrence of dry layers with elevated ozone maximizes between 4 and 6 km.



The global distribution of tropopause folds is rather inhomogeneous with maxima in regions around the jet-streams (e.g., James et al., 2003; Sprenger et al., 2003; Škerlak et al., 2014)). Sprenger et al. (2003) found that the role of the subtropical jet stream (STJ) for STT had been strongly underestimated. The STJ persists during most of the year (Koch et al., 2006) and, thus, could be associated with frequent vertical exchange. However, it

is not only the persistence that matters: high ozone values have also been reported in subtropical intrusions. For example, very strong ozone signatures in the troposphere exceeding 200 ppb have been detected over northern India (Ojha et al., 2014; 2017). High ozone values in the middle and upper troposphere from regions next to the STJ have even been observed above Garmisch-Partenkirchen (Germany) after transport almost all the way around the northern hemisphere (six cases: Trickl et al., 2011). Beyond the regions around the STJ, STT has

been found in the tropics. For example, during 30 % of the MOZAIC flights across the tropical Atlantic at least one event with more than 100 ppb of ozone occurred, which was associated this with strong convection in the Inner-Tropical Convergence Zone (Suhre et al., 1997).

A great surprise was the detection of a substantial stratospheric influence at the mountain-top site Zugspitze (2962 m a.s.l.) in the German Alps and its pronounced positive trend (Fig. 1). In assessments of long-term ozone

series the Zugspitze $O_3$ exhibited the strongest increase (e.g., Parrish et al., 2012; Logan et al., 2012; Oltmans et al., 2012). The Zugspitze ozone increased from 1978 to 2003, in contrast to the neighbouring Wank site (1780 m a.s.l.) where the annual-average ozone level remained constant since the 1980s. We have explained (Trickl et al., 2010) the difference between Zugspitze and Wank ozone by a much larger amount of stratospheric air reaching the higher of the two summits (e.g., Elbern et al., 1997).

The positive trend of stratospheric influence is seen in the contributions of both the "direct" descent to the Zugspitze summit, characterized by clear ozone and humidity structures, and the "indirect" contribution of aged stratospheric intrusions strongly mixed with the surrounding tropospheric air (Fig. 1). The direct contribution (< 10 %) was obtained from correlating low humidity with elevated $^7Be$ (Scheel, 2003); Scheel, pp. 66-71 in (ATMOFAST, 2005)). Aged stratospheric air masses that are completely mixed into the troposphere cannot be

identified by data filtering and, therefore, had not been derived in studies based on observations. In order to obtain some guess of the indirect component of STT in the Zugspitze ozone Scheel (in ATMOFAST, 2005) determined a $^7Be$-to-ozone conversion factor from the direct component, considering that 2/3 of $^7Be$ is produced in the stratosphere (Lal and Peters, 1967). The results in Fig. 1 are associated with an uncertainty of unknown magnitude, in part because of the limited decay time of $^7Be$ (53.42 (±0.01) days; Huh and Liu, 2000). However,

they are plausible since the 1978 stratospheric fraction of 11.3 ppb (31.2 %) is in the expected of values derived from measurement in the late 19[th] century (Volz and Kley, 1988; Marenco et al., 1994). From the decline of the ozone precursors in the 1990s (e.g., Jonson et al., 2006; Vautard et al., 2006) one would expect a negative development. Thus, the estimate of the stratospheric component is perhaps even somewhat conservative (see Tarasick et al., 2019, for a more thorough discussion of possibly higher pre-industrial ozone levels).

A comparable positive ozone trend is also reported for the Swiss Jungfraujoch station (3580 m a.s.l.) where the ozone measurements started in 1992 (Ordoñez et al., 2007). For the lower-lying Italian station Monte Cimone (2165 m a.s.l.) no significant correlation between ozone and the intrusion frequency was detected. However, this conclusion is only based on the years 1996 to 2011, i.e., for a time period that is rather late compared with the ozone rise at the Alpine sites (Cristofanelli et al., 2015). Finally, Colette and Ancellet (2005) retrieved an

increase of stratospheric ozone in the entire troposphere over Europe back to 1970 from the analysis of long-term



ozone-sonde data. Central Europe is located at the exit of the North Atlantic storm track and, therefore, is a receptor region for subsiding stratospheric layers. A possible explanation of the positive trend in our view could be a reaction of the atmospheric dynamics to climate change (see also, Collins et al., 2003; Yang et al., 2016; Lin et al., 2015; Neu et al., 2014).

Dibb et al. (2003) determined the latitudinal distribution of STT over North America by a total of 39 air-chemistry aircraft missions from Boulder (Colorado, U.S.A., 40º N) to up to 86º N. The altitude varied between the surface and about 7000 m. These flights were special since also here [7]Be filter samples were taken as an additional indicator of stratospheric air, an advantage which is normally the privilege of surface sites. Stratospheric influence were identified on 23 of the flights, most of the intrusions being detected near and above 6 km.

Above 6 km the flights took place in stratospheric air during more than half of the time, indicating an extreme transfer of ozone in that altitude range. Below 50 N the stratospheric layers were limited to this range.

The seasonal cycle of the stratospheric ozone contribution at Alpine summit stations exhibits a pronounced summer minimum (Elbern et al, 1997; Stohl et al., 2000; Trickl et al., 2010). In contrast to this, Beekmann et al. (1997) concluded for the entire free troposphere above three European ozone-sonde stations a seasonal cycle

with a slight summer maximum, based on data filtering of ozone profiles between 1969 and 1994. A transition to this behaviour is indicated for growing altitude of the Alpine stations: The summer minimum is least pronounced at the highest of the stations previously compared, Jungfraujoch (see Fig. 1 of Trickl et al., 2010).

Motivated by all these findings, we extend in this paper our STT studies to the full free troposphere. The analysis is based on routine lidar measurements of ozone, water vapour and aerosol since 2007, as well as radiosonde

relative humidity and transport modelling. The resulting stratospheric component in tropospheric ozone over Central Europe is surprisingly strong. We see some confirmation of our findings in recent lidar and sonde results from lower latitudes which will be discussed in detail in Sect. 4.

## 2 Methods

### 2.1 Measurements

#### 2.1.1 IFU ozone DIAL

The tropospheric ozone lidar is operated in Garmisch-Partenkirchen, Germany at IMK-IFU (formerly IFU; 47º 28′ 37” N, 11º 3′ 52” E, 740 m a.s.l.). The laser source is a Raman-shifted KrF laser, and two separate receiving telescopes are used to divide the dynamic range of the backscatter signal of roughly eight decades. This lidar was completed as a two-wavelength differential-absorption lidar (DIAL) in 1990 (Kempfer et al., 1994) and a first

annual sounding series was achieved in 1991 (Carnuth et al., 2002). It was upgraded to a three-wavelength DIAL in 1994 and 1995 (Eisele and Trickl, 1997), leading to a unique vertical range between roughly 0.25 km above the ground and 3 to 5 km above the tropopause, the measurement time interval being just 41 s. By comparing the ozone profiles retrieved from different wavelength combinations (e.g., 277 nm - 313 nm or 292 nm – 313 nm an internal quality check is possible. The choice of an "on" wavelength below 280 nm is particularly beneficial for

achieving a high accuracy and a high vertical resolution (up to about 5 km above the ground). Density has been converted to mixing ratio by using pressure and temperature data from nearby radiosonde stations (Sect. 2.1.4). The noise level of the system since late 2012 is $\pm1\times10^{-6}$ of the input voltage range of the digitizer system. The DIAL features low uncertainties of about $\pm2$ ppb in the lower free troposphere, approximately tripling (under



optimum conditions) in the upper troposphere. Comparisons with the nearby Zugspitze in-situ measurements (at 2962 m a.s.l., see below) show no relevant mutual bias, the standard deviation of the differences being less than 2 ppb. The uncertainty further diminished after another system upgrade in 2012, after introducing a new ground-free input stage to our transient digitizers (Licel) that reduced the noise level by roughly a factor of three. For the

range covered by the near-field receiver (below 1.2 km above the lidar) the uncertainty is of the order of ±5 ppb. The upper-tropospheric performance may be degraded in the presence of high lower-tropospheric ozone concentrations absorbing a lot of the ultraviolet laser emission and by enhanced sky light in summer, in particular in the presence of clouds. The vertical resolution is dynamically varied between 50 m and a few hundred metres, depending on the signal-to-noise ratio decreasing with altitude. Within stratospheric intrusion

layers the vertical smoothing is reduced as far as possible in order to avoid a reduction of the peak concentrations by smoothing. The lidar has been used in numerous atmospheric transport studies (e.g., Eisele et al., 1999; Stohl and Trickl, 1999; Trickl et al., 2003; and other publications cited in this paper).

Aerosol backscatter profiles with very good signal-to-noise ratio up to the lower stratosphere are obtained from the 313-nm "off" channel of the lidar. The methods, implying an ozone correction, have been described by

Eisele and Trickl (2005). Examples demonstrating the data quality achieved in recent years (maximum noise level of the backscatter coefficients: $\pm 1 \times 10^{-7}$ m$^{-1}$ sr$^{-1}$, reached in the tropopause region) can be seen in (Trickl et al., 2015). We derive vertical profiles of the aerosol backscatter coefficients based on on a constant backscatter-to-extinction ratio of 0.020 sr$^{-1}$, which is the average value derived within the European Aerosol Research Lidar Network (EARLINET, 2003). Within clouds larger values are taken, if possible optimized for

minimum discrepancy of the backscatter profiles below and above the cloud.

### 2.1.2 IFU water-vapour DIAL at the Schneefernerhaus high-altitude station

The Zugspitze water-vapour DIAL is operated at the Schneefernerhaus high-altitude research station (UFS, 47° 25′ 00″ N, 10° 58′ 46″ E) at 2675 m a.s.l., about 8.5 km to the south-west of IMK-IFU (Garmisch-Partenkirchen, Germany), and 0.5 km to the south-west of the Zugspitze summit. The full details of this lidar system were

described by Vogelmann and Trickl (2008). It is based on a powerful tunable narrow-band Ti:sapphire laser system with up to 250 mJ energy per pulse operated at about 817 nm and a 0.65-m-diameter Newtonian receiver. Due to these specifications a vertical range up to about 12 km can be reached, almost unaffected by daylight. However, mostly the laser has been operated at half the maximum pulse energy or less to extend the life time of the high-voltage components such as flashlamps. A separation of near-field and far-field signals is achieved by a

combination of a beam splitter and a blade in the far-field channel. The operating range starts below the altitude of the summit station (2962 m a.s.l.). The electronics are almost identical to those of the ozone DIAL. However, at the operating wavelength of 817 nm avalanche photodiodes have been used that introduce higher noise than the photomultiplier tubes preferred for shorter wavelengths. Thus, the system has not yet reached its expected optimum performance in the upper troposphere.

The vertical resolution chosen in the data evaluation is dynamically varied between 50 m in altitude regions with good signal-to-noise ratio and roughly 350 m in the upper troposphere. Free-tropospheric measurements during dry conditions clearly benefit from the elevated site outside or just below the edge of the moist Alpine boundary layer (e.g., Carnuth and Trickl, 2000; Nyeki et al., 2000; Carnuth et al., 2002). After a few years of testing, validating and optimizing the system, routine measurements started in January 2007 with typically two





measurement days per week, provided that the weather conditions are favourable. Operation has been interrupted since winter 2015 due to fatal laser damage. A new Ti:sapphire laser system is under development.

The lidar has been validated in several comparisons with local and remote radiosonde ascents (Vogelmann and Trickl, 2008), an airborne DIAL (Trickl et al., 2016) and the Zugspitze Fourier-transform spectrometer

(Vogelmann et al., 2011). A noise level of 5 % and a bias of 1 % at most was verified to more than 6 km altitude. Furthermore, a very high importance of volume matching in comparisons of water-vapour profiling instruments was found (Vogelmann et al., 2011; 2015), on the scale of a quarter of an hour and a few kilometres.

In some cases, in which a direct comparison of the exact matching of the humidity and aerosol layers was necessary (e.g., Trickl et al., 2016), aerosol backscatter coefficients were retrieved from the "off" wavelength

channel. The calculations were done with a program developed for the IFU aerosol lidar systems (e.g., Trickl et al., 2013; Wandinger et al., 2016).

### 2.1.3 In-situ measurements at the Zugspitze summit and at the Schneefernerhaus station (UFS)

In addition, in-situ data from the monitoring station at the Zugspitze summit (air inlet: 2962 m a.s.l.) have been inspected, namely ozone and relative humidity. Ozone was measured between 1978 and 2012 (e.g., Reiter et al.,

1987; Scheel et al., 1997; Oltmans et al., 2006; 2012; Logan et al., 2012; Parrish et al., 2012). At present the data have been evaluated until 2010. The relative uncertainty of the Zugspitze ozone is 1 %. Ultraviolet absorption instruments have been employed (Thermo Electron Corporation, U.S.A., TE49 analysers). Relative humidity (RH) was registered with a dew-point mirror (Thygan VTP6, Meteolabor, Switzerland) with a quoted uncertainty below 5 % RH. However, the instrument has a wet bias of almost 10 % under very dry conditions (Trickl et al.,

2014). The Zugspitze in-situ measurements were discontinued in January 2013.

After the end of these measurements we have used the corresponding data of the Global Atmosphere Watch (GAW) observatory at the Schneefernerhaus research station (UFS, see $H_2O$ lidar), operated by the German Umweltbundesamt (UBA, i.e., Federal Environmental Agency; 47° 25′ 0″ N, 11° 58′ 46″ E; air inlet at 2670 m a.s.l.). Ozone is continuously measured by ultraviolet (UV) absorption at 254 nm (Thermo Electron Corporation,

model Ts49i). Relative humidity is monitored by the German Weather Service with an EE33 humidity sensor (E+E Elektronik). The calibration of the UBA instrumentation is routinely verified as a part of the GAW quality assurance efforts. The instruments are controlled daily and serviced on all regular work days.

For the comparisons shown in the figures of this paper we use time averages up to one hour because of the time delay of the air mass between UFS and IFU. The comparisons of ozone DIAL and UFS are highly satisfactory,

with differences mostly staying below 2 ppb. However, orographic air-mass lifting must be taken into account that can lead to vertical displacements of ozone structures and larger differences between lidar and in-situ data.

### 2.1.4. Sonde data

Radiosonde data are routinely used for calculating the atmospheric density, which is necessary for quantitative aerosol retrievals and the conversion of the ozone or the water-vapour number density to mixing ratio. Most

importantly, on each measurement day of the ozone DIAL the presence of dry and moist layers was examined in order to identify potential advection from a remote stratosphere or (marine) boundary layer, respectively. The sonde data have been imported from the University of Wyoming data base (http://weather.uwyo.edu/upperair/





sounding.html). Preferentially, the Oberschleißheim ("Munich") sonde RH has been examined, this station (number: 10868) being located 100 km roughly to the north of IFU. If data were not available for a given time or if no indication of an intrusion was found RH profiles from other surrounding stations were used such as Stuttgart (10739, about 200 km to the north-west), Payerne (06610, about 310 km to the west), or Innsbruck

(11120, 32 km to the south, one measurement per day only). In critical cases the station choice was also based on the trajectory results (Sect. 2.2), and even more remote sites have been inspected.

The sonde type used by the German Weather Service (DWD, Deutscher Wetterdienst) during the period presented here was RS 92 (Vaisala; e.g., Miloshevich et al., 2006; Steinbrecht et al., 2008). The sonde data feature an artificial cut-off at 1 % for conditions when the UFS DIAL revealed even even much drier conditions

(Trickl et al., 2014).

### 2.2 Transport modelling

#### 2.2.1 LAGRANTO

Four-day forward trajectories have been calculated since September 2000 once a day for start times $t_0 = 1:00$ CET (Central European Time = UTC + 1 h), $t_0 + 12$ h, $t_0 + 14$ h and $t_0 + 36$ h based on the Lagrangian Analysis

Tool (LAGRANTO; Wernli and Davies, 1997a; Sprenger and Wernli, 2015; http://www.lagranto.ethz.ch). On each day, trajectories are calculated using operational forecast data from the European Centre for Medium-Range Weather Forecasts (ECMWF) interpolated to a longitude-latitude grid with 1°×1° horizontal resolution. With respect to the vertical levels, ECMWF uses 137 vertically hybrid levels since June 2013 (and 91 levels before), where 24 levels are between 250 and 600 mbar. For each start time the four-day forward trajectories are calculated

starting in the entire region covering the Atlantic Ocean and Western Europe (20º east to 80º west and 40º to 80º north) between 250 and 600 mbar. From this large set of trajectories those initially residing in the stratosphere (potential vorticity larger than 2.0 pvu) and descending during the following four days by more than 300 mbar into the troposphere were selected as "stratospheric intrusion trajectories". The same selection criterion was used in a previous case study (Wernli, 1997b) to study an intrusion associated with a major North Atlantic cyclone.

Since June 2001 so-called "intrusion hit tables" have been additionally distributed that crudely estimate how stratospheric air develops over several days as a function of altitude above the four STACCATO (Stohl et al., 2003) partner stations Jungfraujoch, Zugspitze, Monte Cimone and Thessaloniki. Both the STT trajectories and the hit tables are distributed daily to all interested partners and institutions. Intrusion warnings based on these images have been issued by IFU if several of the stations could be affected (Zanis et al., 2003b).

For special case studies LAGRANTO has been operated with re-analyses meteorological data, for periods up to five days (e.g., Trickl, 2014; 2016). The three-dimensional wind fields for the calculation of the trajectories were taken from the ERA-Interim data set (Dee et al., 2011), which was interpolated to a 1°×1° horizontal grid and provides winds at 6-h intervals. The number of vertical levels in ERA-Interim is 60, with 11 levels between 250 and 600 mbar.

#### 2.2.2 HYSPLIT

For analysing intrusion events with travel times exceeding the four days set in the operational LAGRANTO forecast runs or with source regions outside the domain of the forecasts we use HYSPLIT (Hybrid Single-



Particle Lagrangian Integrated Trajectory, Draxler and Hess, 1998; Stein et al., 2015; https://ready.arl.noaa.gov/ HYSPLIT.php) backward trajectories. HYSPLIT is easy to operate on the internet and allows one to perform analyses of the vertical profiles with an adequate expenditure of time in an intense programme of vertical sounding. We mostly used the standard version with three trajectories initiated at different altitudes within or

close to a layer of interest, selecting the option "model vertical velocity". These trajectories are extended over the maximum 315 h. If necessary, several of these runs were started with slightly varying initial conditions. Multi-trajectory ensembles have also been used to create some "backward plume" (Trickl et al., 2013). However, these ensembles did not cover a sufficient number of days and have been applied just in a few case studies. If one or more trajectory bundles did not reach an altitude range typical of the lower stratosphere in the outflow

region of an intrusion (e.g., roughly 7.5 km or more in boreal regions) within 315 h the case was rejected. Just in few cases extension trajectories were calculated to verify the stratospheric source (Trickl et al., 2015). For some case studies also the FLEXPART model has been used with a time span of 20 days (Trickl et al., 2014). FLEXPART produces a much more complex output beyond the requirements of the current study with thousands of measurements.

For many years we preferentially selected "reanalysis" meteorological data. Although the re-analysis data are coarser than other meteorological data available they have led to a superior model performance in the free troposphere in many of our studies (Trickl et al., 2010; 2013; 2014; Fromm et al., 2010) and the analysis of our routine measurements. Despite the known limitations of backward trajectories (e.g., Stohl and Seibert, 1998) most specific free-tropospheric ozone layers in years of observations could be related to reasonable sources with

this operation mode of HYSPLIT, the best investigated transport type being STT. Since 2014 near-real-time data evaluation and aerosol archiving in the EARLINET (European Aerosol Research Lidar Network, https://www. earlinet.org) data base have been achieved. Thus, GDAS-based trajectories (GDAS: Global Data Assimilation System) have been taken since the re-analysis-based model version are available only with considerable delay. The re-analysis mode was applied later on just if a GDAS run did not verify STT.

Slight vertical displacements of intrusion layers at the northern rim of the Alps exist in the model runs as reported previously (Trickl et al., 2010; 2015). These offsets, that vary from case to case, are explained by the insufficiently resolved orography that leads to an altitude of IMK-IFU (730 m a.s.l.) roughly half-way between the valley (Garmisch-Partenkirchen) and the Zugspitze summit. It is important to activate the check box "terrain" on the HYSPLIT input page together with the altitude option "AMSL" (above mean sea level). In this case, the

absolute height is used on the vertical axis as well as the contour of mountains are displayed, and better agreement with the altitude of an arriving atmospheric layer is achieved. The trajectories reproduce air-mass lifting above mountain ranges, which is particularly spectacular above Greenland with a surface altitude of about 3 km maintained over hundreds of kilometres.

## 3 Results

### 3.1 Description of the data analysis and interpretation

Starting in 2007, routine measurements have been started with both DIAL systems. This has yielded vertical profiles of ozone, water vapour and aerosol backscatter coefficients, derived from the 313-nm channel of the ozone DIAL. The number of measurements is particularly high in the case of the ozone lidar, resulting in a total



of 2275 evaluated data files between 2007 and 2016 (Table 1). The present study is, therefore, based on the ozone profiles during this period and all other profile data are used for identifying the source for conspicuous ozone structures such as stratospheric air intrusions. Measurements have been made on a large number of fair-weather days or during short periods of clearing. However, really strong efforts to make at least one measurement were limited to the EARLINET (European Aerosol Research Lidar Network) "climatology days" Monday and Thursday (EARLINET, 2003). Ancillary information from sondes and trajectories has been gathered for each measurement day.

There are several gaps in the data of the ozone DIAL. These gaps are explained by extended periods of laser or computer damage, sometimes involving the search for new technical solutions for the system. The latest one occurred between August 2016 and September 2017.

Ozone is not always a good tracer of STT. Particularly in winter frequently very small exceedances of the background ozone level are found in intrusion layers, related to departure from the lowermost layer of the stratosphere. As a consequence, we decided to use elevated ozone just as a secondary indicator of STT (Trickl et al., 2010), low humidity being the strongest (Trickl et al., 2014; 2015; 2016).

For identifying stratospheric intrusions, layers with ozone exceeding the neighbouring background by at least 10 % were analysed with LAGRANTO forward and/or with HYSPLIT backward trajectories. For HYSPLIT, potential vorticity was not available for identifying tropopause crossings and, thus, descent of the trajectories from roughly 7.5 km or more (preferentially at latitudes higher than 50º N) was applied as criterion to diagnose a stratospheric layer. This altitude had been found to be sufficient for identifying STT from previous analyses (e.g., Trickl et al., 2010) and from the LAGRANTO calculations that identify stratospheric trajectories based on potential vorticity. At lower latitudes and in summer higher start altitudes (9-12 km) have been chosen. In addition, we require low lidar or sonde relative humidity (RH) with minimum values $\leq 10$ % to co-exist at least at adjacent altitudes. In most cases the minimum RH was clearly below this threshold. In confirmation of our results from the water-vapour DIAL (Trickl et al., 2014; 2015; 2016) we found typical minimum RH values of 1-2 % in the sonde data for source regions over the North Atlantic or neighbouring regions (intrusion Types 1-5 as defined by Trickl et al. (2010) mostly by distinguishing source regions), 1 % being the lowest value found in the sonde listings (Sect. 2.1.4).

For the most frequent long-range descent from a remote stratospheric source (e.g., central or western Canada, Alaska, Siberia: Type 6 (Trickl et al., 2010)) or slow descent from the North Atlantic minimum RH mostly ranged between 3 and 6 %. Interestingly, the reverse also holds: finding RH values in this range in sonde data very reliably point to very long transport times. Quite surprisingly, the longest descent analysed (15-17 days) led to negligibly low $H_2O$ in the DIAL measurements at UFS (July 16, 2013; Trickl, 2015).

Intrusions reaching altitudes around 3 km were verified by looking at the Zugspitze ozone and RH data until 2010 and the UFS data afterwards. As pointed out in Sect. 2.1.3 the Zugspitze RH rarely dropped clearly below 10 %.

## 3.2 Typical findings

As previously discussed (Trickl et al., 2010) stratospheric air intrusions passing over Garmisch-Partenkirchen arrive from almost all directions. Easterly directions mostly result from detours of the dry layers via Eastern Europe or curl formation over Central Europe potentially in cut-off lows.





Intrusion layers can be observed under many different conditions. We routinely observe prefrontal and post-frontal intrusion layers, as well as intrusions slowly descending from the far west. The prefrontal cases are frequently associated with stratospheric air masses descending from the Arctic to North Africa or the Mediterranean basin followed by some return flow to Central Europe. These layers normally rise as they are on a transition into a warm conveyor belt (e.g., Cooper et al., 2004). Postfrontal intrusions mostly reach low altitudes above Garmisch-Partenkirchen and occur after virtually all cold fronts, of course also in the "classical" case of beginning anti-cyclonic conditions (e.g., Stohl and Trickl, 1999; Trickl et al., 2003). They can, however, also occur between two frontal passages that are sometimes separated by not more than one day. In these cases the inclined descending layer can be sandwiched between the low-lying clouds of the preceding front and the high-lying clouds of the incoming new front.

A few specific remarks:

*(1) Very intense intrusions have been rare*

Although intrusions with 100-150 ppb of ozone in the middle and upper troposphere are not that rare, much higher values are really exceptional. Just three cases with peak ozone mixing ratios reaching or exceeding 200 ppb have been found during the period described here (2007-2016). The most intense intrusion, also covering a wide vertical range, was observed on 1 October 2015 (Fig. 2). The peak ozone mixing ratio on that day was 235 ppb and rapidly dropped to less than 100 ppb. For comparison we give average values of the in-situ GAW measurements at UFS (2670 m). The UFS ozone data exhibit a slight negative bias of 2 to 4 ppb in the morning. This bias is outside typical differences between the lidar and UFS or Zugspitze summit and is ascribed to orographic lifting of the air masses that arrived from the east (see below), along the former glacier basin.

The presence of an intrusion on 1 October 2015 had been predicted by the LAGRANTO operational forecasts. For this paper the trajectory calculations were repeated with ERA Interim re-analysis data and extended from four to five days. In Fig. 3, we give one example for a start time of 12 UTC (13 CET) on 25 September. This start time is slightly too early with respect to the lidar observations, but the trajectories clearly show a stratospheric intrusion passing over Garmisch-Partenkirchen. At later times, the main trajectory structures persist, but the trajectory become more and more complex with streamers for quite different arrival times over Central Europe.

Figure 3 and the trajectory plots for later times show high-lying and low-lying trajectories over Southern Germany and qualitatively confirm the observations in Fig. 2. Some more clarity comes from the HYSPLIT backward trajectories that allow one to select start times and altitudes above the lidar. The HYSPLIT calculations started at altitudes where high ozone was observed (not shown) nicely confirm the main pathway shown in Fig. 3 with a northward departure over Greenland followed by a decent via Eastern Europe.

Further exceptional mixing ratios observed on 26 February 2015 (235 ppb), and, in a particularly spectacular case (Trickl et al., 2014) on March 6, 2008 (200 ppb).

*(2) Extremely thin layers can survive the long-range transport with almost negligible mixing*

The width of intrusion layers can vary considerably from case to case. Layer widths clearly exceeding 2 km, in particular that in Fig. 2, are not frequent, in agreement with the analysis of Colette and Ancellet (2005).



Also very thin layers with widths of down to 0.2 km have been observed. Both IFU DIAL systems are capable of resolving these structures, and there is mostly very little mixing with tropospheric air (Trickl et al., 2014; 2015; 2016). A particularly spectacular case (26-27 December 2008) of a thin, very dry (RH = 1 %) layer was discussed by Trickl et al. (2014) and verified by high-vertical-resolution FLEXPART transport modelling. Here, we show as an example an even more exciting case from December 2013 of two parallel very thin high-ozone layers descending to Alpine summit levels (Fig. 4). Again, the minimum RH was 1 %, the cut-off level of the sonde (Trickl et al., 2014).

*(3) Slow long-distance descent (Type 6) dominates the observations above about 4.5 km*

The slow descent of stratospheric layers from remote source regions such as Western Canada, Alaska or even Siberia down to Alpine summit levels was identified by Trickl et al. (2010). These Type-6 intrusions were observed much more frequently above 4.5 km than at the Zugspitze summit. A particularly spectacular case on 16 July 2016 was analysed by Trickl et al. (2015): The trajectories indicated a descent from the stratosphere above Siberia over roughly two weeks without a resolvable rise in humidity. A source of STT can also be the subtropical jet stream over Asia, reaching mid- and high latitudes over the Pacific Ocean (Trickl et al., 2011).

*(4) Intrusion layers frequently arrive via North Africa*

Long fronts reaching as far as the Saharan desert are typically associated with the advection of dust (Papayannis et al., 2008). Prefrontal intrusion layers have been mostly located above the dust layer (exceptions exist). In Fig. 5 we show time series of the ozone aerosol profiles on 31 January and 1 February 2014, and Fig. 6 displays three HYSPLIT trajectories selected for the three relevant ozone and aerosol layers at midnight between the two days. The dust was lifted to roughly 5 km which is typical of these Föhn events at the northern rim of the Alps (e.g., Jäger et al., 1988; Papayannis et al., 2008). Above the dust layers a layer with elevated ozone passed over Garmisch-Partenkirchen. The minimum Munich RH on 1 February at 1 CET was 2 % indicating a moderate travel time (the UFS DIAL was not operated). The intrusion trajectory in Fig. 6 shows rather rapid transport from about 10 km above Cape Farvel (Greenland) to North Africa. Obviously, this high speed makes it possible to pass eastward over the southern part of the front and enter the air stream rising to the Alps. The HYSPLIT trajectories for other altitudes around 7 km start to differ strongly vertically and horizontally upstream of Eastern Canada. The LAGRANTO STT forecast confirms at least subsidence from Labrador to North Africa.

Another dust case is shown in Figs. 7 and 8 for 18 June 2013, this time combined with long-range descent from the Northern United States (U.S.) (and presumably beyond) and a single-loop curl at low latitudes, with a corresponding trajectory analysis in Fig. 8. Here, the intrusion air mass crossed the cold front over Western Spain or Portugal. At 13 CET on 18 June this front extended north-south from Bristol (U.K.) to North Africa. Obviously, the upper end of the clouds was rather low in this area, similar to the May-1996 case (Trickl et al., 2003). The RH determined from a comparison of the results of the water-vapour DIAL (right panel in Fig. 7) and the Munich radiosonde was 8 to 12 %. These rather high values are ascribed to the very long descent during at least thirteen days.

Intrusions co-existing with Saharan dust were observed on a total of 67 days. The number of dust days in our record is limited because frequently dust arrives below clouds, which impedes lidar measurements.


*(5) Summer-time ozone in the middle and upper troposphere has been mostly high*

Elevated ozone was a feature observed many times above about 4.5 km during the warm season. Two examples from May and August 2015 are shown in Fig. 9. In both cases (and most others) dry layers exist within the high-ozone range and the corresponding HYSPLIT trajectories stay at high altitudes. For the altitudes in these examples analysed with trajectories we did not find any contact with potentially polluted planetary boundary layer (PBL) within the maximum of 315 h provided by the model. However, the calculation for a cirrus layer around 7.5 km on 10 August ended around 5 km over the Pacific, giving some hint on the origin of the moisture required for the cloud formation.

*(6) Volcanic and fire aerosol are transported downward from the lowermost stratosphere*

During the periods of major volcanic activity impacting the lower stratosphere frequently aerosol was detected in intrusion layers (see also Browell et al., 1987). More specifically, particles in intrusion layers were registered after the eruptions of Okmok and Kasatochi (July and August 2008, respectively; see (Trickl et al., 2016)), Redoubt (March 2009), Sarychev (June 2009) and Nabro (June 2011) (more details: Trickl et al., 2013). Typical 313-nm aerosol backscatter coefficients were $5 \times 10^{-7}$ m$^{-1}$ sr$^{-1}$ and less. The highest value was $2.35 \times 10^{-6}$ m$^{-1}$ sr$^{-1}$, observed on 7 September 2009 after the violent eruption of Sarychev. Stratospheric intrusions have been identified as a highly important mechanism for the rapid depletion of stratospheric aerosol in the mid-latitudes within one year or less (Deshler, 2008; Trickl et al., 2013). This includes strong fires (pyro-Cbs) that normally just reach the lowermost stratosphere (e.g., Fromm et al., 2008; 2010), the latest presumable smoke case being 2 October 2017 (not shown).

*(7) Dry air layers can also arrive from the lower-latitude Atlantic*

Quite a few of the dry layers with elevated ozone have been traced back to the south-west of the Azores Islands. Here, the trajectories frequently form curls or spirals exiting backward towards the north-west (e.g., Fig. 8). In several cases extension trajectories were calculated and confirm descent from high altitude at high latitudes. Only these verified cases were accepted as "stratospheric", given the low humidity.

*(8) Intrusions rarely penetrate into the PBL*

Some intrusions have been observed to slide along the top of the PBL over several days without a clear indication of penetration into it. This suggests that descent towards the ground is likely to occur mostly during night-time. This could explain why Reiter et al. (1990), based on years of ozone soundings with the Eibsee-Zugspitze cable car (1.0 km to 2.95 km a.s.l.), did not observe any case of subsidence to below 1.4-1.6 km a.s.l.: the cable car runs only during day-time. By contrast, Eisele et al. (1999) reported a case of sufficiently deep early-morning descent of a STT layer that it could be caught by the forming PBL. Similar conclusions are reported by Ott et al. (2016). We have not studied this topic systematically because of missing local water-vapour data below 3.0 km.

**3.3 STT and long-range transport of boundary-layer ozone**

The typical ozone rise to sometimes even more than 100 ppb in the middle and upper troposphere during the warm season mentioned in Item 5 of Sect. 3.2 is remarkable. Since such a behaviour is mostly absent during the darker period of the year this could suggest that a higher ozone background due to strong photochemical activity is imported from remote regions such as North America or East Asia. An analysis carried out in 2005 based on



just eight-day FLEXTRA trajectories yielded North American influence over Garmisch-Partenkirchen during already 28 % of the time for the period 4/2003-9/2004 (ATMOFAST, 2005).

Since we did not analyse vertical ranges with moderate (or constant) ozone for the period since 2007 this could suggest that the air quality in the United States (U.S.) has improved. Nevertheless, we did identify several cases of intercontinental transport of pronounced amounts of ozone. One of these cases is described below. In general, a clear distinction of long-range advection from the stratosphere or a polluted PBL is a complex task beyond the scope of this paper. It requires more suitable analysis tools such as the FLEXPART model as applied by us in detail in previous studies (e.g., Trickl et al., 2003; 2010; 2011). There, we extended FLEXPART-based analyses to up to 20 days and found merging air streams from the stratosphere and remote boundary layers (Trickl et al., 2011).

It is very difficult to estimate the stratospheric contribution in the summertime middle and upper troposphere. The sonde data show very low humidity mostly in rather confined layers and rarely in the entire range with elevated ozone. This can indicate mixing in merging air streams, but could also be attributed to differences in vertical distribution with respect to the rather remote radiosonde stations. In the case of mixing an assessment of the stratospheric component would have to rely on model-based estimates which can be rather crude (e.g., Trickl et al., 2014).

As one example of a significant North American ozone plume we describe here the case of 28 May 2015. On that day a major part of the summertime ozone step could be related to a high-ozone episode in the Eastern U.S. Again, a part of the step (the lowest section) is related to STT. The ozone profiles from that day are displayed in Fig. 10. At almost all tropospheric altitudes there were (in part remarkable) changes in ozone that could be explained by the RH and trajectory analyses. In the morning two intrusion layers from source regions around Alaska are discernible at about 4.7 and 3.1 km (RH = 6 % and 4 %, respectively). The trajectories seem to continue rising for backward times beyond −315 h, to altitudes higher than 7 km. These intrusions diminish later on.

In the afternoon an ozone step to roughly 80 ppb formed above 6.7 km (light blue, red and grey curves in Fig. 10). The air mass was rather humid (Munich sonde RH 50-76 % at 13:00 CET; some backscatter profiles showed signal from cirrus clouds), with the exception of the lowest peak for which the Munich radiosonde yielded RH = 4 % (there at 7.6 km). Figure 11 shows three 315-h HYSPLIT backward trajectories selected for relevant altitudes above our site (7400 m, 8200 m, 8500 m) from a larger number of trajectories calculated. The ozone rise near 7 km corresponds to long-range descent from northern Alaska, in agreement with the low RH.

The higher trajectories bend southward over the Great Lakes and follow the Mississippi back to the Caribbean Sea. On 24 May an altitude of 1.5 km is reached, i.e., above Louisiana. The strong air-mass rise from the Gulf of Mexico to Canada suggests the presence of a warm conveyor belt. Indeed, we verified the presence of a rather wide warm conveyor belt with tools described by Madonna et al. (2014) and Sprenger et al. (2017). To the east a wide zone with peak ozone exceeding 80 ppb is revealed in Fig. 12 indicating long-lasting high pressure. The trajectories propagate along the west side of that zone and indicate some overlap with the high-ozone region. This result is highly satisfactory and confirms our excellent experience with HYSPLIT run with the re-analysis option. In fact, we also tested the GDAS option that offers better spatial resolution. As in earlier comparisons (e.g., Trickl et al., 2016), we found strong deviations from Fig. 11, with almost all trajectories leading backward to Alaska at high altitudes, approximately parallel to the red 7400-m trajectory in Fig. 11. This result is in





considerable disagreement with the RH data that show a narrow dry layer just above 7 km and elevated humidity at higher altitudes that is in excellent agreement with the import from the Caribbean Sea suggested in Fig. 11 and the cirrus signal in the lidar backscatter data.

**3.4 Statistical analysis of STT**

The ozone profiles alone do not allow us to quantify free-tropospheric ozone budget due to STT. The boundaries of the intrusion layers cannot always be clearly distinguished. Water-vapour profiles show a clearer contrast, but rarely match in time or space for a meaningful comparison with ozone. The soundings with the $H_2O$ DIAL were not always made on the same day, and sonde data could only be used as a qualitative tool due to the long distance to the stations.

As a consequence we decided to perform a statistical analysis of the fraction of the measurement days per month with one ore more identified intrusion layers (Sec. 3.1), similarly to the approach by Beekmann et al. (1997; see Sect. 4). The seasonal cycle from our analysis is shown in Fig. 13, based on all (585) measurement days for each month between 2007 and 2016. The month-to-month variability of the fractions is rather low. If we derive a standard deviation from months with at least eight measurement days the overall standard deviation was 0.12,

which looks unrealistically high considering the smoothness of the data in Fig. 13.

As seen from a comparison with the fractions derived by including all measurements days, the variability is much higher when taking just the data for Monday and Thursday, i.e., the EARLINET "climatology days" (a total of 286). Nevertheless, the principal course of the seasonal cycle is retained which proofs the absence of a significant bias due to the selection of the measurements days. Just for the period December to February the

deviations are larger due to a smaller number of climatology days covered.

Similarly to Beekmann et al. (1997) we find a rather weak seasonal cycle of the fraction of intrusion days. However, our average fraction of 84.1 % is much higher than that in their study (4.8 % for Uccle, Belgium). We discuss this fact in Sect. 4. A slight seasonal maximum is visible in Fig. 13 around August, resembling the findings by Beekmann et al. (1997) for the observational criteria used in their study.

Considering the pronounced summer minima of deep STT obtained for the Alpine summit stations Jungfraujoch (3580 m), Zugspitze (2962 m), Sonnblick (3106 m) and the station Mte. Cimone (2165 m) in the North Italian Apennines (Stohl et al., 2000; Trickl et al., 2010) there must be a pronounced free-tropospheric summer maximum of the STT fraction to explain the overall seasonal cycle with a slight maximum in August. The overall seasonal cycle in Fig. 13 is dominated by Type-6 intrusions (Trickl et al., 2010), i.e., intrusions that

originate in source regions far to the west including East Siberia. Most of these intrusions descend to 4.5 to 5 km at most and are, thus, missed at the Alpine stations.

For comparison with the Zugspitze results, we provide in Fig. 13 the fractions of days on which 3.0 km was reached. These fractions agree rather with those from the analysis of the Zugspitze in-situ data for 2001-2005 in (Trickl et al., 2010), shown in Fig. 13 as "TT2010". The distributions agree rather well. However, the lidar-based

fractions are rather noisy due to a moderate number of measurements days. This may have influenced in particular the deviations from the Zugspitze analysis for January and March. However, since 2008 rather cold winters have prevailed and perhaps could have led to a weaker influence of the North Atlantic storm track and less frequent deep intrusions in January. The March maximum in the Zugspitze fraction was caused by an





elevated count of six and four deep-intrusion days in March 2014 and 2015, respectively. These were the only years with a reasonable number of measurement days in March which could suggest the occurrence of a positive outlier. On the other hand, the FLEXPART simulation in Fig. 1 of Trickl et al. (2010), made for 1995-1999, shows a pronounced maximum in March.

The peak ozone mixing ratio in intrusion layers can vary considerably from case to case. High ozone values beyond 100 ppb are mostly limited to altitudes above 5 km. However, very recently (on 3 November 2017) we found as much as 95 ppb of ozone in a layer centred at 3.0 km and featuring a full width at half maximum of just 0.2 km. In order to see if there is a systematic seasonal variation we determined the seasonal cycle of the "weak" and "strong" intrusions in the free troposphere (Fig. 14). The intrusions for which the peak ozone values

exceeded the neighbouring ozone "background" by less than 15 ppb exhibit a summer minimum, whereas those with an exceedance of more than 40 ppb a summer maximum, with some contribution also from January to April. Trickl et al. (2014; 2016) hypothesized that the peak ozone concentration is related to how far the intrusion layer extends into the stratosphere when the descent starts. In this view the initial layer thickness would be most pronounced in summer.

Lidar observations are restricted to fair-weather conditions. Thus, the fractions derived in this study are likely to exhibit some fair-weather bias. To estimate the ozone import from the stratosphere we, therefore, selected the year 2014, when 70 % of the EARLINET climatology days were covered by measurements. 2014 was also the only year with a high number measurements made in all months. The same high fraction of intrusion days (more than 80 %) was found as for the entire period. The missing 30 % of climatology days were almost exclusively

due to measurements prevented by bad weather. If one assumes that no intrusion was present on bad-weather days (which is not true according to the model-based predictions), the overall fraction sinks from 84.1 % to 59 % which is, still, remarkably high. We derived for 2014 a very crude estimate of the average fraction of directly detectable stratospheric ozone in the free troposphere by considering its peak structure. This potentially conservative average fraction of 18 % (varying from roughly 12 % in winter to roughly 26 % in summer), was

obtained from determining for all days the ozone mixing ratio crudely integrated over all identified co-existing intrusion layers in comparison to the integrated ozone outside these layers up to the thermal tropopause on a given day. The width of a layer could frequently just be guessed.

This analysis cannot assess the role of stratospheric ozone fully mixed into the troposphere since this "indirect" contribution does not exhibit discernible ozone peaks and deep humidity minima. This indirect contribution

cannot be estimated without model assistance. For the Zugspitze summit an estimate of the indirect component was made based on the [7]Be measurements (Fig. 1) that lead to an overall annual stratospheric ozone fraction of the order of 40 % for the first years after 2000.

## 4 Discussion and Conclusions

The lidar measurements of tropospheric ozone at IMK-IFU have reached a performance that makes possible to

resolve ozone structures with amplitudes of 5 to 10 ppb up to the middle troposphere. This is particularly important during the cold season when a considerable number of intrusions with weak excess ozone occur (Fig. 14). Also in the upper troposphere, covered by the less sensitive wavelength pair 292 nm − 313 nm, considerable improvements have been achieved and structures with amplitudes of 10 to 15 ppb can be resolved. This is



important although intrusion peaks are usually more pronounced in this altitude range. Together with the humidity and trajectory analysis, it has allowed us to identify a high number of the stratospheric layers in the free troposphere.

The very large fraction of stratospheric intrusion days in our lidar measurements is an enormous surprise. The observed average percentage of 84.1 % of all measurement days in the free troposphere since 2007 strongly exceeds the average fraction of about 17 % observed at the neighbouring Zugspitze summit (2962 m) between 2001 and 2005 (Trickl et al., 2010). It strongly exceeds the 27 % of temporal coverage of the measurements by tropopause folds determined from lidar measurements at the Table Mountain Facility in Southern California between 2000 and 2015 (Granados-Muñoz and Leblanc, 2016), located at much lower latitude (34º N). However, the Table Mountain seasonal cycle of the stratospheric fractions peaks at about 70 % in winter and, during that period, almost matches our fraction. From July to September it minimizes at 1 to 2 %, in qualitative agreement with the jet-stream minimum over the western United States during that period (Koch et al., 2006).

STT analyses have also been derived from ozone-sonde data (e.g., Beekmann et al., 1997; Colette and Ancellet, 2005; Kuang et al., 2017; Tarasick et al., 2019). Ozone sondes yield a substantially lower temporal data coverage than lidar sounding. However, sondes offer the advantage of simultaneous measurements of ozone and RH as well as the absence of a fair-weather bias. The results obtained from the sondes have been rather controversial which seems to be the consequence of the individual data-filtering approach. Recent sonde studies confirm a rather high percentage of STT: At the same latitude as Table Mountain, but in the south-eastern United States, ozone-sonde measurements at Huntsville (Alabama, U.S.A., 35º N) were analysed between May and September 2013 (Kuang et al., 2017). There, the summer minimum was 23 % (August), and 50-60 % were reached in May-June. A presumable winter maximum could, again, lead to even higher fractions. Tarasick et al. (2019), in a series of sonde campaigns in Canada between 2005 and 2007, observed STT events every two to three days in spring and summer and every four to five days in autumn and winter. As in our observations the stratospheric influence maximized in the middle and upper troposphere with average 34 % (22 ppb) of directly identified stratospheric ozone. Colette and Ancellet (2005) made a thorough study for European stations back to 1970 based on about 27000 profiles. They concentrated on the statistics of individual intrusion layers and found a pronounced summer maximum of the ozone content of the layers.

By contrast, Beekmann et al. (1997) derived a much lower tropopause-fold fraction per year of just 5 % from European sonde data for different periods around 1990. However, apart from this low fraction their work confirms the missing summer minimum for the free troposphere as a whole above our site.

This low fold fraction obtained by Beekmann et al. (1997) is puzzling. However, the data selection criteria in the 1997 assessment were much stricter than in the present study and in that by Granados-Muñoz and Leblanc (2016). The selection of cases by the observational groups in the 1997 effort was based on an ozone increase by at least 25 %, a potential-temperature gradient of more than 11.5 K/100 mbar, RH < 25 % and a wind speed of more than 20 m s$^{-1}$. In addition, the vicinity of an upper tropospheric jet stream and co-incidence with an upper-level front was required. Apart from RH, where our approach is much stricter, all other conditions thresholds are likely to reduce the fraction, and the product of all the partial fractions could, indeed, be rather small. The additional criteria could exclude many of the aged intrusions that originate over rather remote regions and dominate our observations in the middle and upper troposphere. For example, the trajectory results for Type-6





intrusions frequently yield a wind speed of the order of 1000 km in 24 h (11.6 m s$^{-1}$) during the approach to Central Europe. One additional possibility is that the analysis of Beekmann et al. (1997) focussed just on isolated peaks which could exclude some of the wider structures we observed in the middle and upper troposphere. In addition, a potential increase in STT frequency between the early 1990s and recent years must be considered.

However, the Zugspitze in-situ results (Fig. 1), which reveal most of the increase in deep STT occurred between the 1970s and 2000, suggest that such a contribution is less important.

In contrast to this, as mentioned in the introduction, Dibb et al. (2003) determined a huge fraction of stratospheric ozone from a large number of aircraft missions between Colorado and the Arctic Sea, up to more than 85%. The analysis involved $^{7}$Be as a tracer.

In summary, high stratospheric fractions have been reported for several stations and airborne measurements at different latitudes though not all around the year as in our data. This, together with other findings cited in the Introduction indicate a much higher importance of STT than thought in work more than two decades ago. Similar analyses for significantly more stations at different latitudes are desirable. Some of the long-term sonde series such as Uccle, Hohenpeißenberg and Payerne should be analysed again with revised data-filtering criteria

in order to make the results more comparable with more recent work.

The absence of a pronounced summer minimum in the free-tropospheric seasonal cycle in Fig. 13 as compared to the minimum in the Alpine summit-station data (Stohl et al., 2000; Fig. 1 in (Trickl et al., 2010)) suggests the existence of a summer maximum of transport pathways leading to the derived seasonal cycle. In addition to the summer maximum of the fractions (Fig. 13) Fig. 14 suggests a summer maximum of the ozone content in free-

tropospheric intrusion layers, perhaps due to thicker layers separating from the lowermost stratosphere during the warm season. In fact, Trickl et al. (2014) concluded that most intrusions originate in the lowest layer above the dynamical tropopause. This was confirmed in the same paper by the almost negligible drop in Zugspitze CO in STT layers and was verified by LAGRANTO transport modelling of the LUAMI (Lindenberg upper-Air Methods Intercomparison) case (Trickl et al., 2016).

However, a robust estimate of the summertime ozone transfer from the stratosphere to the troposphere from our data is difficult since not all of the strongly elevated ozone above 4.5 km correlates with low humidity: The temporal overlap with the measurements with the water-vapour DIAL until the 2015 laser damage was frequently not sufficient, and the dry layers (RH ≤ 10 %) observed above the rather remote radiosonde stations rarely cover the full range of elevated ozone.

Diagnosing the elevated ozone above about 4.5 km in summer (Figs. 9, 10), were also water vapour is highly variable (Vogelmann et al., 2015), requires a much more elaborate approach. Outside the dry layers quite frequently the source regions could not be identified due to missing ascent of the trajectories from the PBL within 315 h (HYSPLIT maximum) or no clear indication of descent from the stratosphere. Trickl et al. (2011), focussing on just six cases found the occurrence of high-ozone upper-tropospheric air streams that originated

from merging ascending and descending air masses along the subtropical jet stream. That study included calculations over fifteen to twenty days with the FLEXPART model. Such a considerable effort has been beyond the scope of the current investigation based on an intense programme of routine measurements extended over many years. Just a crude estimate was made for a single year (2014) indicating a summertime stratospheric fraction of free-tropospheric ozone of the order of 26 %. This fraction does not include the longer-lived





"indirect" stratospheric component that, due to mixing, cannot be identified from the observations available such as in the case of Fig. 1.

Certainly, improved modelling will be needed to quantify STT. So far, Eulerian models have had difficulties in reproducing the strong ozone rise at the Alpine sites (e.g., Parrish et al., 2014; Staehelin et al., 2017). The calculated ozone rise reported by Parrish et al. (2014) and Staehelin et al. (2017) ends almost 20 years earlier than the observed one. In Eulerian models higher spatial resolution is needed for reproducing deep STT (Roelofs et al., 2003; Trickl et al., 2010; Rastigejev et al., 2010; Eastman and Jacob, 2017) as well as reduced free-tropospheric mixing (Trickl et al., 2014). Due to the limited free-tropospheric mixing Lagrangian approaches look promising.

In any case, transport modelling must be extended to about 20 days. This study has revealed that the transport pattern of the intrusions is dominated by slow descent from Canada, Alaska and Siberia (Type 6 as defined by Trickl et al. (2010)). The trajectories frequently exhibit horizontally wavelike transport paths, but mostly without strong vertical variation. This kind of long-range descent, its underlying dynamics and its influence on the STT budget call for a meteorological explanation. It would also be interesting to determine how much an extension of the transport calculations to at least fifteen days (as suggested by our analyses) would change the STT budget with respect to earlier work (e.g., Sprenger et al., 2003; Škerlak et al., 2014).

A great hope is the planned re-analysis of the Zugspitze ozone series by using the refined data-filtering criteria derived by Trickl et al. (2010). Here, additional information is available from the [7]Be measurements, and the in-situ data do not exhibit a fair-weather bias. However, such an effort must also account for the source conditions: The tropopause region is a mixture of about 50% stratospheric and tropospheric air each (Shapiro, 1980; Vogel et al., 2011). The stratospheric portion of the descending air mass can vary significantly, also depending on the stratospheric residence time (Reiter et al., 1975). We find in our trajectory studies that not all air parcels stay in the lower stratosphere for a long period of time: Sometimes air masses descending to the Alps had stayed above the tropopause for less than a single day, after a pronounced rise even from low altitudes. All these facts also mean a challenge in future modelling efforts.

**5 Data availability**

The data can be obtained on request from the authors of this paper (thomas.trickl@kit.edu, hannes.vogelmann@kit.de; ludwig.ries@uba.de, Michael.sprenger@env.ethz.ch). The 313-nm aerosol back-scatter coefficients are archived in the EARLINET data base, accessible through the ACTRIS data portal http://actris.nilu.no/.

**6 Author statement**

TT and HV carried out the lidar measurements. LR made available the in-situ data for the Schneefernerhaus station. MS provided the operational daily STT forecasts with LAGRANTO as well as selected model runs based on re-analysis meteorological data. TT interpreted the observations and prepared the manuscript, assisted by the co-authors.

**7 Competing interests**

The authors declare that they have no conflict of interest.





**Acknowledgements**

The authors thank H. P. Schmid for his support. We are indebted to H. E. Scheel who detected the importance of STT in the Zugspitze ozone in the late 1990s and set the motivation for our STT studies ever since. Due to his early dead in 2013 he can no longer co-author this paper. They acknowledge the assistance by H. Giehl and M.

5 Perfahl as well as the great support by the UFS team. W. Steinbrecht provided ozone sonde data of the German Weather Service (DWD) for comparisons. B. Pierce helped us with an ozone analysis for the U.S. The development of the Zugspitze water-vapour DIAL has been funded by the Bavarian Ministry of Economics and German Bundesministerium für Bildung und Forschung within the programme Atmosphärenforschung 2000 (ATMOFAST project: Atmospheric Long-range Transport and its Impact on the Trace-gas Composition in the

10 Free Troposphere over Central Europe, (ATMOFAST, 2005)). The aerosol observations contribute to EARLINET (European Aerosol Research Lidar Network, currently partly founded by ACTRIS 2).

The service charges for this open access publication have been covered by a Research Centre of the Helmholtz Association.



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



**Table 1.** Measurement days (first line for a given year) and number of evaluated measurements for a given month (second line); **sumd** means the the number of measurement days, **summ** the number of evaluated measurements

| Year | Jan | Feb | Mar | Apr | May | June | July | Aug | Sep | Oct | Nov | Dec |
|------|-----|-----|-----|-----|-----|------|------|-----|-----|-----|-----|-----|
| **2007** | | | | 8 | 8 | 6 | | | 2 | 8 | 7 | 7 |
| | | | | 21 | 50 | 15 | | | 7 | 28 | 22 | 16 |
| **2008** | | 4 | 3 | 7 | 11 | 5 | 1 | 7 | 2 | 9 | 12 | 5 |
| | | 14 | 25 | 28 | 41 | 9 | 5 | 24 | 11 | 28 | 67 | 45 |
| **2009** | 8 | 10 | 1 | 9 | 1 | 4 | 9 | 11 | 3 | 7 | | |
| | 45 | 63 | 1 | 46 | 2 | 12 | 49 | 56 | 19 | 51 | | |
| **2011** | | | | | | | 4 | 1 | 5 | 6 | 6 | 1 |
| | | | | | | | 12 | 6 | 16 | 16 | 21 | 3 |
| **2012** | | | | | | | | | | | 1 | 2 |
| | | | | | | | | | | | 6 | 4 |
| **2013** | 2 | 6 | 2 | | | 5 | 16 | 2 | 11 | 13 | 6 | 15 |
| | 3 | 21 | 8 | | | 20 | 52 | 9 | 42 | 60 | 21 | 70 |
| **2014** | 9 | 16 | 20 | 10 | 8 | 18 | 9 | 8 | 9 | 14 | 13 | 12 |
| | 37 | 59 | 64 | 39 | 22 | 43 | 27 | 24 | 28 | 34 | 40 | 36 |
| **2015** | 13 | 16 | 12 | 18 | 7 | 17 | 4 | 16 | 11 | 12 | 11 | |
| | 37 | 61 | 51 | 63 | 26 | 51 | 11 | 57 | 31 | 52 | 52 | |
| **2016** | | | | | 2 | 10 | 11 | | | | | |
| | | | | | 4 | 54 | 45 | | | | | |
| **Sumd** | 33 | 51 | 38 | 52 | 37 | 65 | 54 | 45 | 43 | 69 | 56 | 42 |
| **Summ** | 122 | 218 | 149 | 197 | 145 | 204 | 201 | 176 | 154 | 269 | 229 | 174 |

**Total number of measurements days:**     **585**
**Total number of evaluated measurements:**    **2238**



**Figures:**

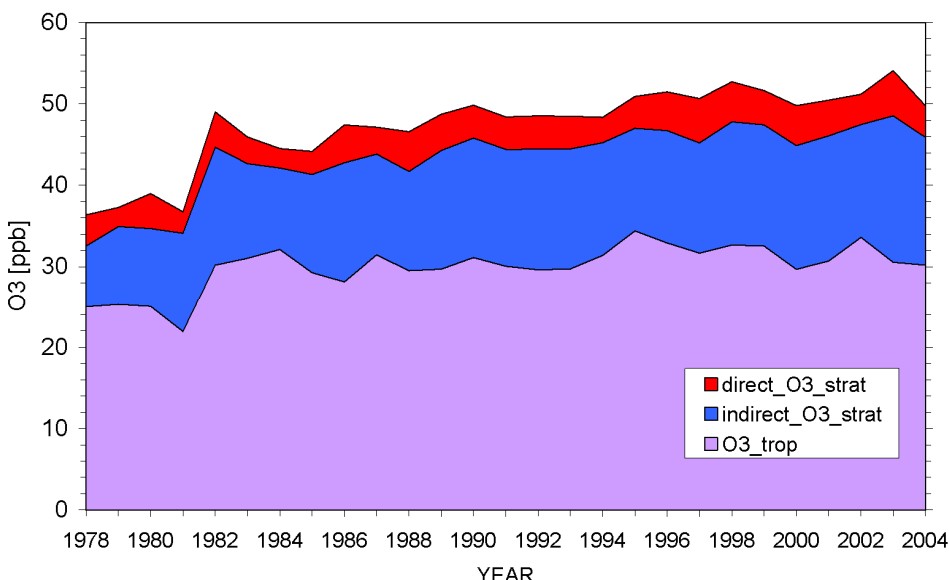

**Fig. 1.** Annual mean ozone mixing ratios for the Zugspitze summit from 1978 to 2003, together with preliminary estimates of the directly detected stratospheric component (red) and of the indirect component (blue) obtained from [7]Be measurements: The stratospheric influence doubled during that period. As a consequence after 1981 the positive ozone trend disappears after subtracting the evaluated stratospheric fraction of ozone. The figure of H. E. Scheel is taken from the ATMOFAST final report (2005; Fig. 2.40 on p. 67).

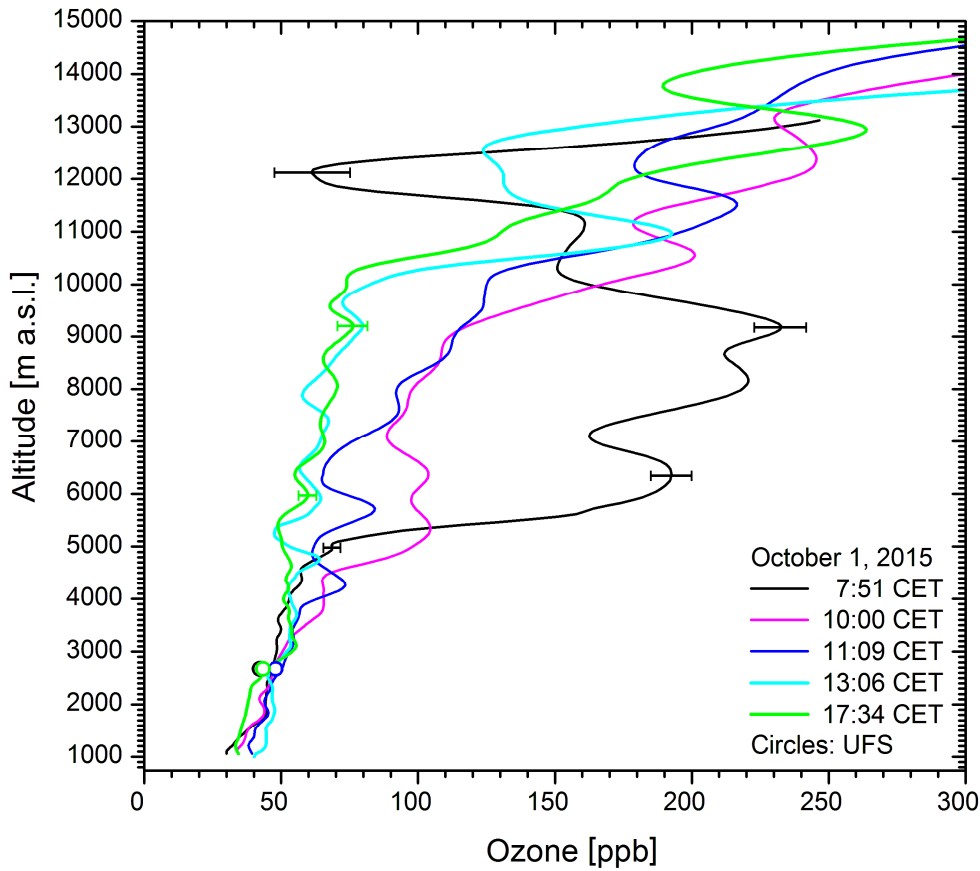

**Fig. 2.** Vertical ozone profiles from the lidar measurements on 1 October 2015; 235 ppb is the highest mixing ratio ever measured with the IFU DIAL since the beginning of the measurements in 1991. The distribution changes dramatically within about ten hours. The Munich thermal tropopause level was 10454 m (0 UTC = 1 CET) and 11903 m (12 UTC). The minimum sonde RH was 2 % (0 UTC) and 1 % (cut-off level, 12 UTC). The in-situ data (1-h averages) of UFS (2670 m) are marked with circles coded in the same colours as the lidar measurements next to the same time. A few error estimates representative for the respective altitudes are given for a judgement of the data quality.

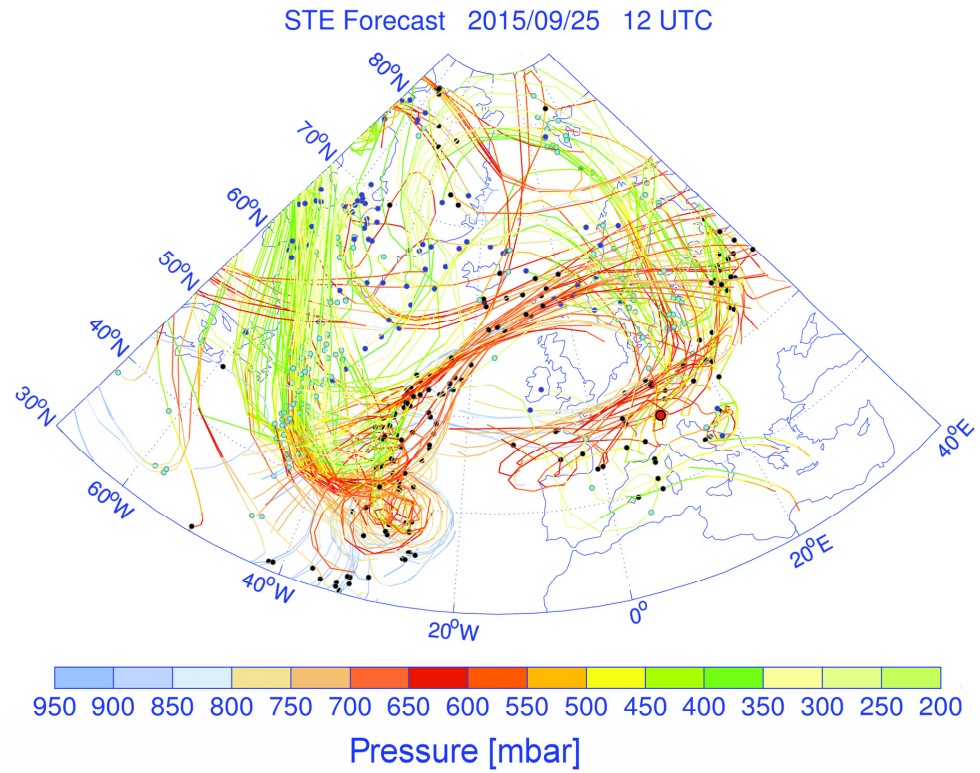

**Fig. 3.** Recalculation of the LAGRANTO forward trajectories based on ERA-Interim wind data: The length of the trajectories is five days. Less than 1 % of the trajectories are displayed for clearness. The pressure level is colour coded in mbar. The start time $t_0$ is 25 September 2017, 12 UTC (13 CET), marked with dark blue dots.

5     The times $t_0 + 2$ d and $t_0 + 4$ d are marked with bright blue and black dots, respectively. The red dot north of the Italian peninsula marks the position of Garmisch-Partenkirchen.





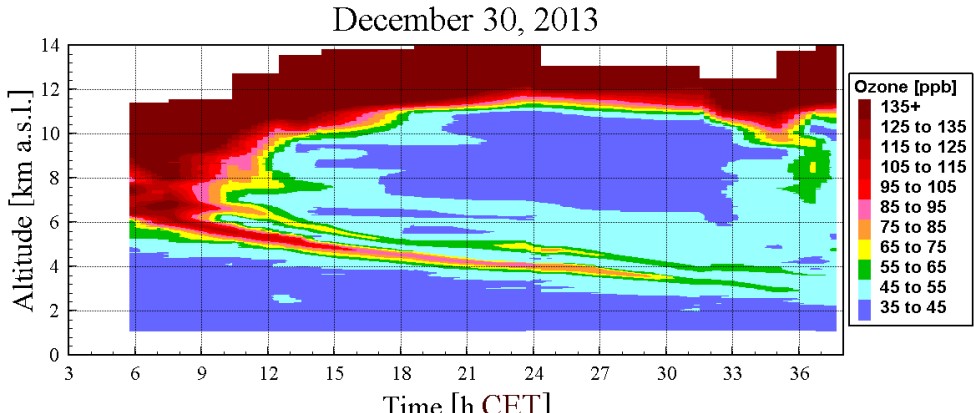

**Fig. 4.** DIAL ozone soundings on 30-31 December 2013 showing two narrow layers descending parallel to below 4 km a.s.l.

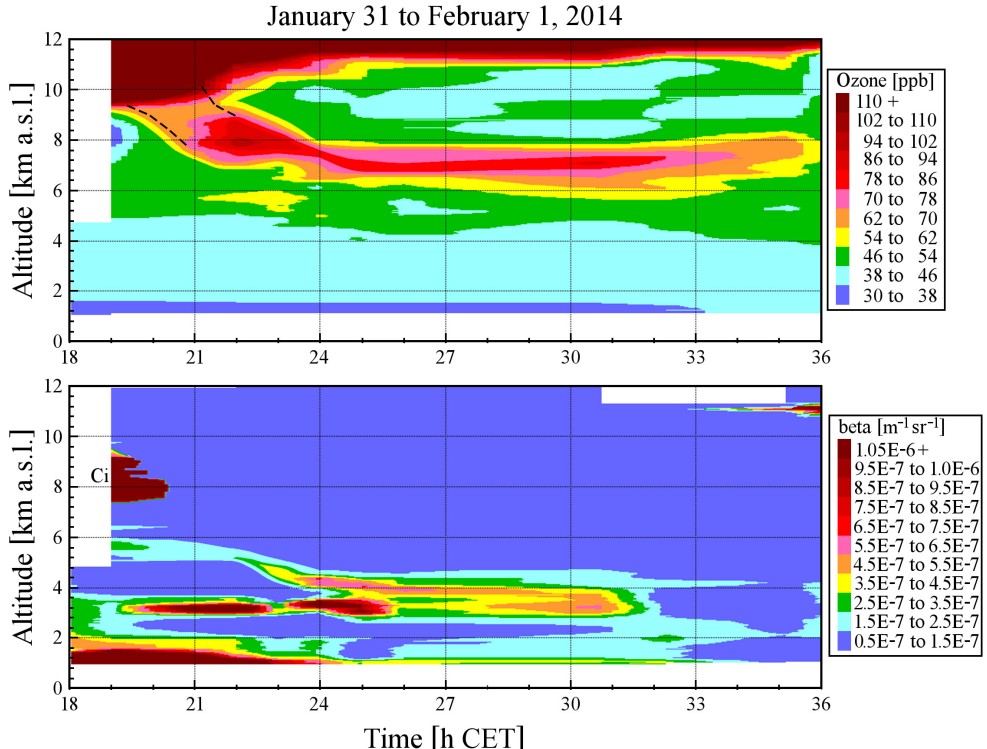

**Fig. 5.** Time series of ozone (upper panel) and the 313-nm aerosol backscatter coefficient (lower panel) on 31 January and 1 February 2014: Both a stratospheric intrusion layer (6.5-9 km) and a weak to moderate Saharan dust event 2.5-6 km are seen. Due to a 3-h data gap between 19 and 22 CET the colour coding during this period is highly uncertain. The intrusion could also have formed a full tropopause fold as indicated by the dashed black lines.



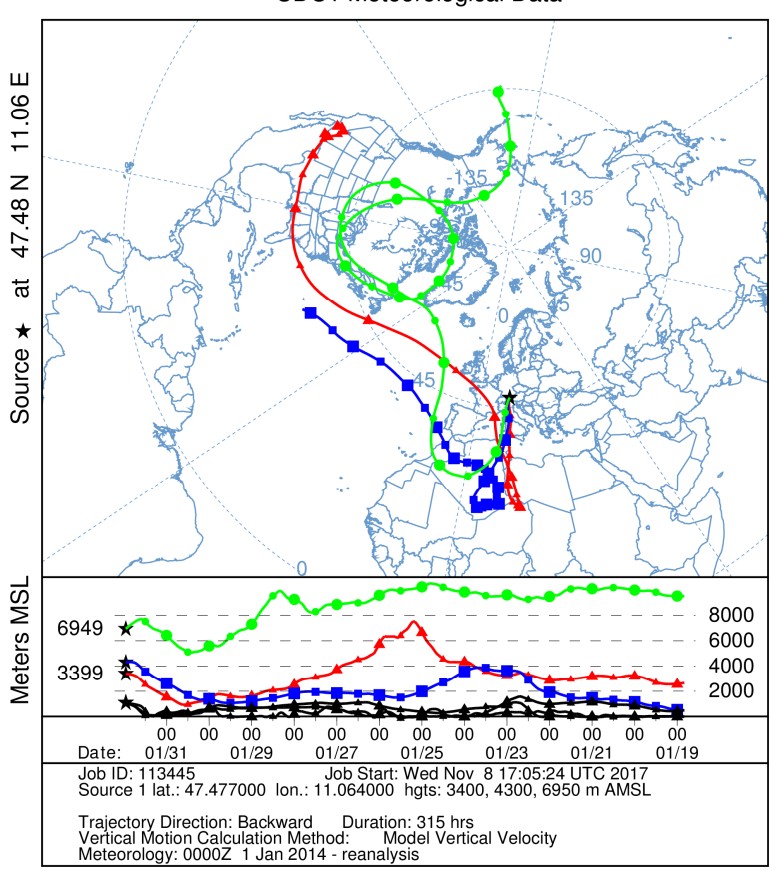

**Fig. 6.** Selected HYSPLIT 315-h backward trajectories starting above Garmisch-Partenkirchen on 1 February 2014 at 0:00 CET within the two aerosol layers in Fig. 5 (lower panel) between 3 and 4.5 km as well as in the stratospheric intrusion between 6 and 9 km (upper panel in Fig. 5); the black curves at the bottom of the vertical cross section mark the ground levels for each trajectory.



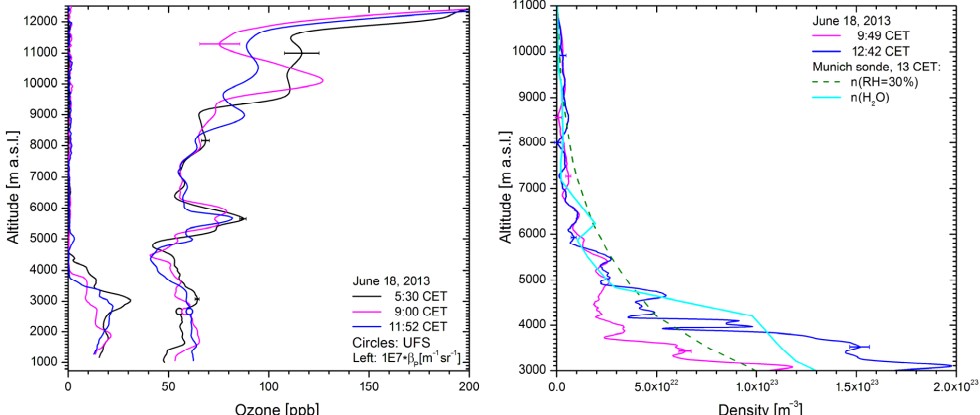

**Fig. 7.** Profiles of ozone, the aerosol backscatter coefficient (left panel) and the water-vapour number density n (right panel) from DIAL measurements at IFU and UFS on 18 June 2013 during a Saharan dust event; for comparison the densities for RH = 30 % and the measured RH by the Munich radiosonde are shown. The stratospheric intrusion peak at around 5.7 km corresponds to 8-12 % RH because of long-range descent (Fig. 8). The times specified for the UFS DIAL are the end times of the respective measurement (lasting about 16 min). The in-situ data of UFS (2670 m) are marked with circles coded in the same colours as the lidar measurements next to the same time and confirm the values from the lidar measurements within 1-2 ppb.





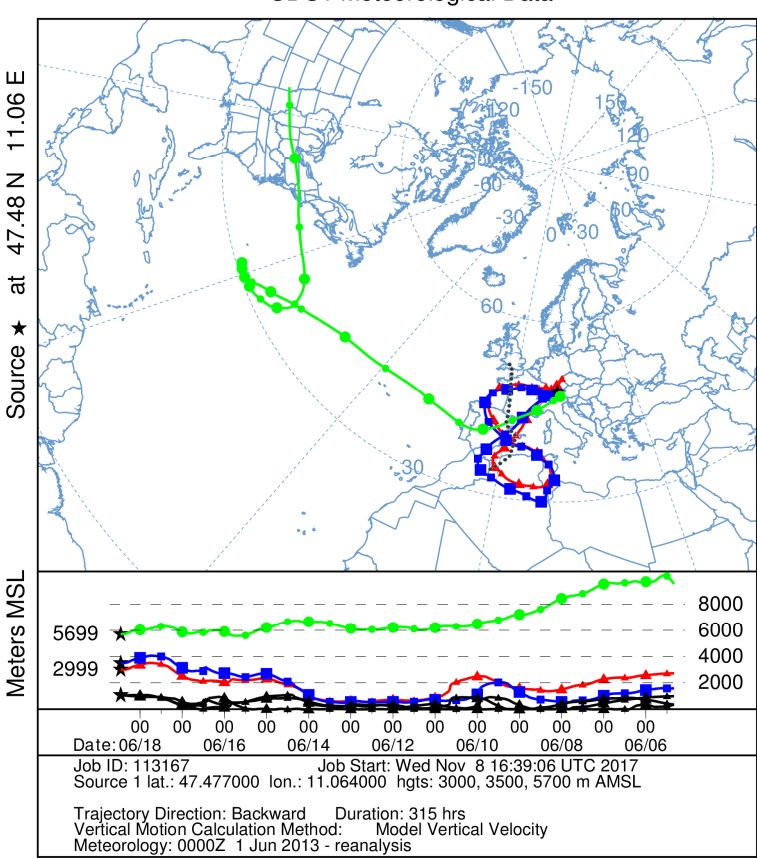

**Fig. 8.** Selected HYSPLIT 315-h backward trajectories starting above Garmisch-Partenkirchen on 18 June 2013 at 11 UTC (12 CET) within the Saharan dust layer as well as in stratospheric intrusion at 5.7 km (see Fig. 5); the black curves at the bottom of the vertical cross section mark the ground levels for each trajectory. The dotted black line indicates the arriving cold front on 18 June 2013, 13 CET.



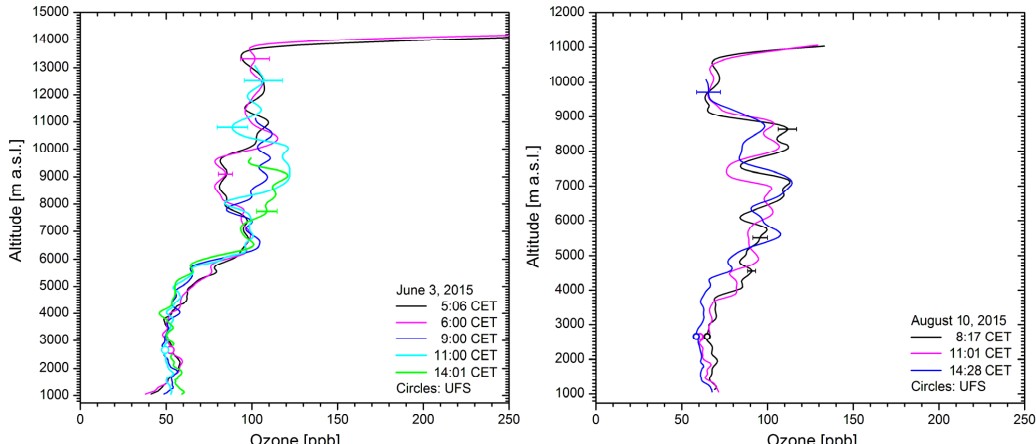

**Fig. 9.** Ozone profiles from 3 June 2015 (left panel) and 10 August 2015 (right panel) showing very high ozone above 5 km and 3.8 km, respectively; the high values are mostly explained by STT. The in-situ data (1-h averages) of UFS (2670 m) are marked with circles coded in the same colours as the lidar measurements next to the same time and confirm the values from the lidar measurements within 1 to 2 ppb. Munich tropopause: 13782 m (3 June, 0 UTC), 13764 m (3 June, 12 UTC), 12471 m (10 August, 0 UTC) and 11029 m (10 August, 12 UTC).

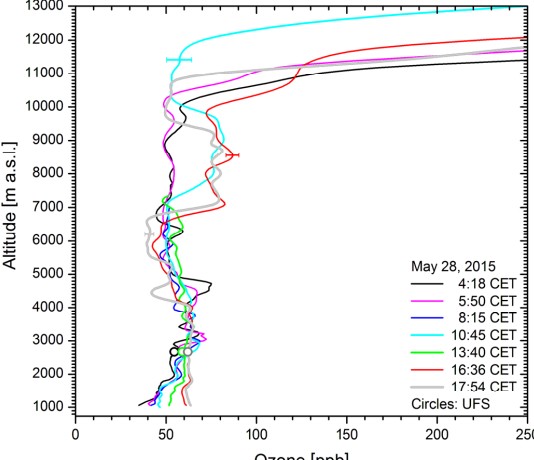

**Fig. 10.** Profiles of ozone from DIAL measurements at IFU (Garmisch-Partenkirchen) on 28 May 2015; the pronounced diurnal cycle of the vertical distribution is ascribed to a quick change in source regions at different altitudes. Between 7.5 and 9.5 km about 80 ppb from a high-ozone area in the U.S. east of the Mississippi was observed. Intrusion layers are temporarily seen at about 3.1 km, 4.7 km and 7.1 km. The in-situ data of UFS (2670 m) are marked with circles coded in the same colours as the lidar measurements next to the same time. The temporary upward shift of the tropopause (bright blue curve) was caused by a humid layer containing a cirrus cloud.


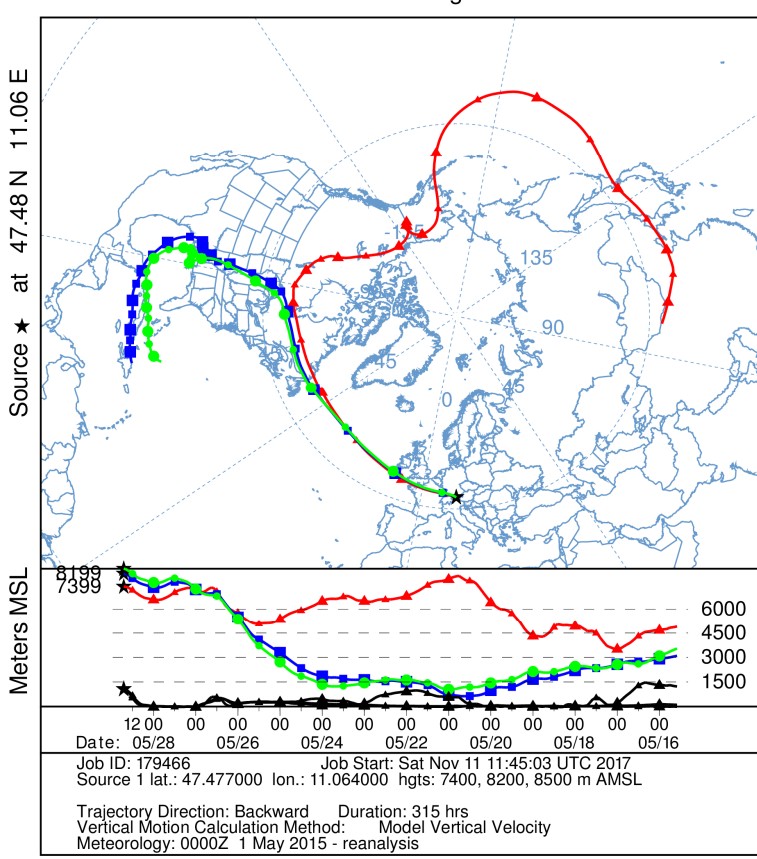

**Fig. 11.** Selected HYSPLIT 315-h backward trajectories starting above Garmisch-Partenkirchen on 28 June 2013 at 17 UTC (18 CET) for three relevant altitudes in the high-ozone ozone range of Fig. 12; the black curves at the bottom of the vertical cross section mark the ground levels for each trajectory. The mountains on 5/22 (Alaska) and 5/16 (China) belong to the red trajectory.



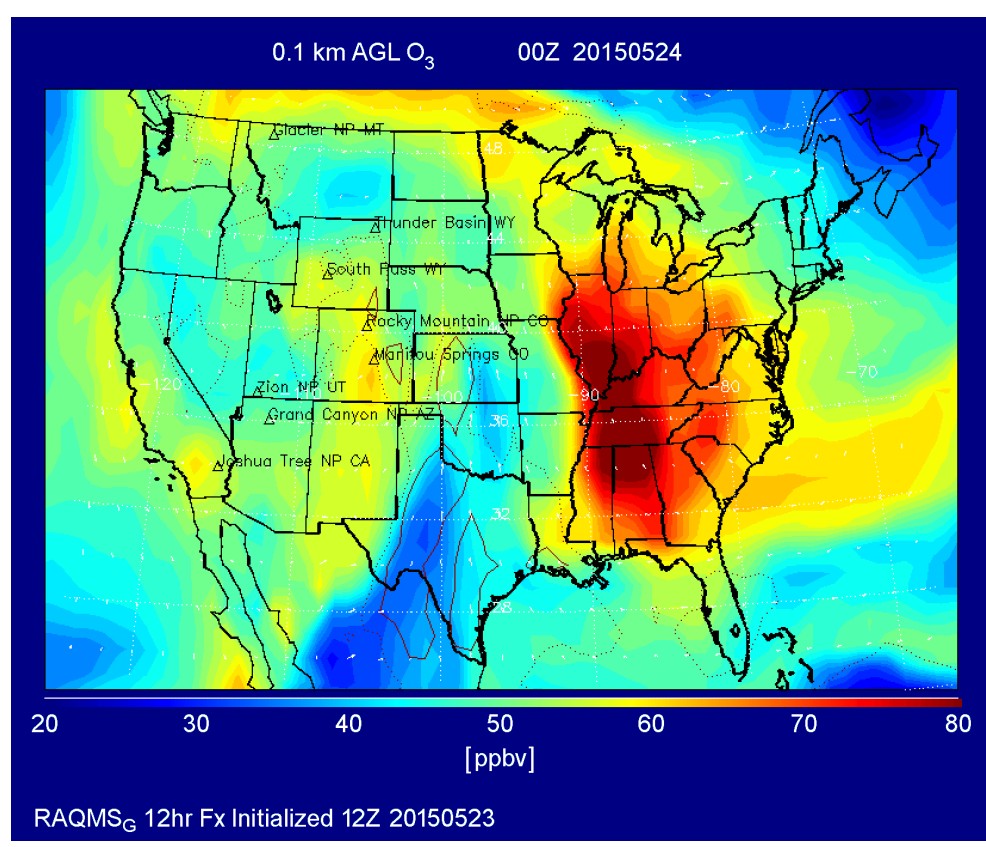

**Fig. 12.** Ozone over the U.S. on 24 May 2015 showing mixing ratios of more than 80 ppb east of the Mississippi
(RAQMS (Real-time Air Quality Modeling System) calculation kindly provided by B. Pierce (NOAA))

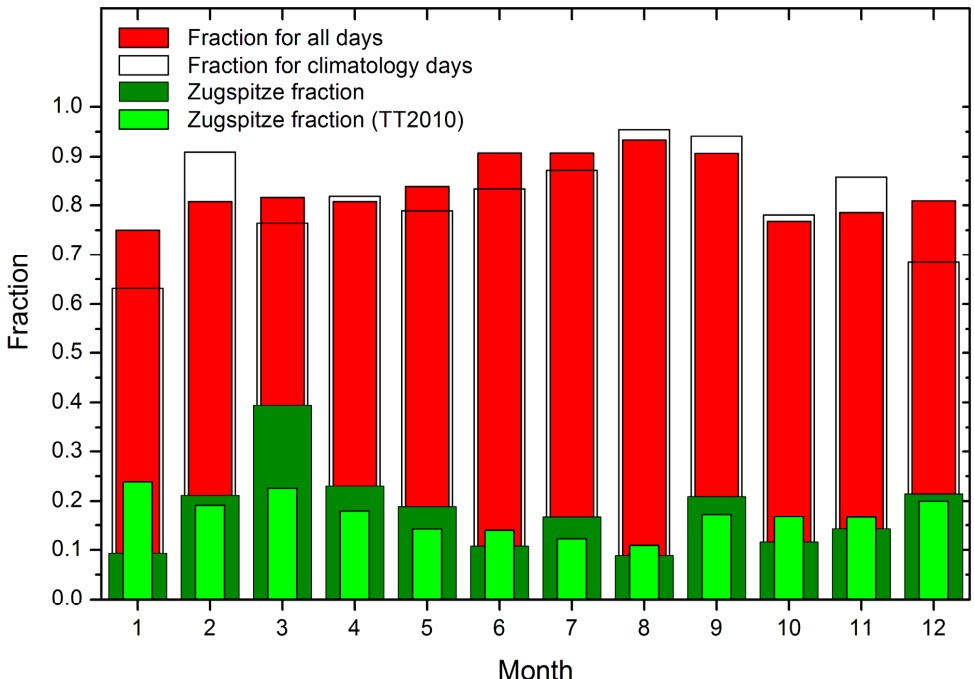

**Fig. 13.** Fraction of intrusion days in the ozone lidar data averaged for each month between 2007 and 2016 (see text); we give the fraction for all measurement days (red bars) and for the "climatology days" Monday and Thursday (transparent bars). For comparison, the same analysis is shown for those intrusion days that show stratospheric influence at 2962 m (Zugspitze, dark green), together with the maximum fractions calculated from the Zugspitze in-situ data underlying Fig. 12 in (Trickl et al., 2010, "TT2010"; light green).



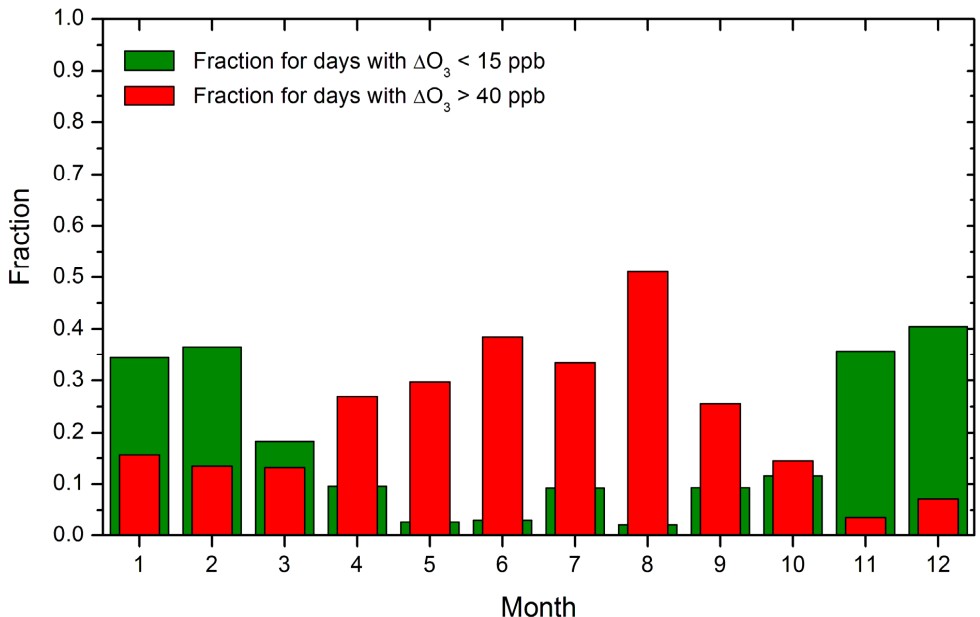

**Fig. 14.** Fraction of intrusion days in the ozone lidar data with weak and strong ozone peaks averaged for each month between 2007 and 2016 (see text)