# Peer review of "Very high stratospheric influence observed in the free troposphere over the Northern Alps – just a local phenomenon?"

_Atmospheric Chemistry and Physics, 2019_

## Referee Comment (RC1) · Anonymous Referee #1 · 25 Sep 2019

The paper deals with the analysis of a 9-year lidar data base (ozone UV DIAL and water vapour) to discuss the statistical occurrence of stratospheric intrusion above Southern Germany. The objectives are to extend previous estimate of the fraction of intrusion days derived from mountain top observation at 2962 m. Indeed the lidar data base appears well suited for such a task. Interesting results are discussed in section 3.4 and Fig. 13 and 14, and they deserved to be published. However the paper is not very well written with many digressions not necessary to discuss the results of the statistical analysis. The section 3.1 which is critical to understand the methodology, is not very clear and a table with the list of criteria to identify stratospheric intrusions must be provided. It is hard to see if the water vapour lidar is actually used for the

relative humidity assessment. Section 3.2 and 3.3 are either very qualitative or out of scope and could be easily shortened or removed. Overall the paper must be published but with a major revision of the presentation. There are also many self citation to previous works which are not always necessary. Finally it is a pity a similar approach is not applied to the Hohenpeissenberg ozonesonde data base to alleviate error due to ozone/humidity layer mismatch and to obtain a complementary statistical study of intrusions in the same region.

Specific comments

p1 l37: Tarasick et al. 2019 (accepted for publication in Elementa) recently show that the 10-ppb figure from early ozone observations is strongly underestimated by a factor of 2. This will increase the estimate of transport from the stratosphere when using such a number. Please acknowledge this likely underestimate in the paper

p2 l10: add Ebel et al. for regional model studies https://doi.org/10.1016/S1352-2310(97)00063-0

p2 l29Âă: Colette et al. 2006 https://doi.org/10.1029/2006GL025793 also analyzed the lifetime and mixing of laminae in the free troposphere. Add citation

p3 l5: add Kowol et al. 2000 https://doi.org/10.1029/2000GL011369d where ozone exchange near the STJ is quantified.

P3.l32: O3 precursor did not decline everywhere. Ozone still increases in Eastern Asia (see Parrish et al. ACP 2012)

p3.l22: Better to show a figure with the Zugspitze data up to 2013 for a better comparison with the lidar time period 2007-2016. Considering the rise of tropospheric ozone in Fig. 1 (30→46ppb), the direct ozone intrusion is actually decreasing from 1978 to 2004 and only the indirect contribution is increasing form 1/3 to $\frac{1}{2}$.

P4.L18-22: After a nice overview of stratospheric ozone transport at different latitude and altitude, one would expect a better description of the actual goals of this work in

the introduction and how the authors will fill the gaps identified in the introduction.

p.6 l13-20 Only half of the lidar record corresponds to the in-situ data set. How do the authors deal with this ?

p.6 l33to37 Do you mean that RH from radiosonde are used in conjunction with the O3 lidar to identify intrusion ? What is the uncertainty due to layer mismatch or radiosonde poor sensitivity in very dry layers ? Why are the Hohenpeissenberg ozonesonde not used in the analysis of your case studies ?

p.7 l25-30 Is a paragraph advertising for availability of the "intrusion hit tables" really necessary in this paper ? Better to discuss here Fig. 3 which is a good example of a LAGRANTO output.

P8. l15-18 The author cannot claim the re-analysis are better than operational analysis without a thorough discussion about the methodology to compare the two meteorological fields. The following sentence about validation of backtrajectories with observations of ozone is good enough to justify the use of HYSPLIT in this paper. Is the diabatic air mass vertical motion in the UTLS properly accounted for when using the analyzed vertical wind in 15-day trajectory calculation ?

p.9 This section is quite important because it is the backbone of the statistical analysis. It deserves a summary table to list the different criteria to identify a stratospheric intrusion: ozone and RH thresholds, from which instruments, need for forward of backward trajectories. The discussion about the humidity values within stratospheric intrusion is hard to follow. I understand a 10% RH threshold is taken. But the following discussion focuses on a list of case studies with smaller RH values. Please rewrite the paragraph l24 to 34 (emphasize that 10 % is good enough and avoid multiple references to your past studies). It is not clear if the aerosol lidar data are actually used and how. If not discussion about the aerosol lidar in section 2.1 is unnecessary and the lengthy presentation of Saharan dust or volcanic/fire aerosol transport events is not needed in section 3.2.

p.9 l.1 Better to emphasize the number of ozone measurements days (585) which is a better proxy to assess how meaningful is the statistical study. For example in Fig. 10 on May 28 there are only two clearly independent profiles while 7 are reported. Give also the number of water vapour lidar observations coincident with the ozone data.

p.9.L13 What do you mean by ozone as a secondary indicator ?

P9.L15 A 10% ozone increase seems a very low threshold considering first the measurement accuracy (5%-10%) and second the ozone variability due other transport processes (horizontal differential advection, convection).

Section 3.2 and 3.3 These two sections are way too long and is of little value for the analysis developed in section 3.4. I recommend to keep only quantitative information needed for the statistical study e.g. two examples of ozone vertical profiles: a thin and thick intrusion and the ancillary information to identify the stratospheric origin. Quantitative information of some features discussion section 3.2 can be indeed developed: - very intense intrusion are rare (define and give occurrence fraction) - thin layers are frequent (same recomm.) - slow descent are frequent above 5 km (same recomm.) - intrusion does not penetrates in PBL (how many cases ?) Discussions about the dust cases or the volcanic episode are out of scope and do not add a new perspective in the analysis. I get the feeling the authors wish to emphasize the lidar capabilities of monitoring both aerosol and ozone without making an actual use of this capability for a better assessment of the stratospheric intrusion frequency in Southern Germany. P.14 l.5-10 Do you mean that the water vapour lidar is not used ? Please clarify how you truly derive the RH needed for the stratospheric intrusion analysis. This is a critical parameter and large bias can occur if the two measurements are not co-localized.

p.14 l16-20 Please explain what you are doing here. Why do you focus on data taken on monday and thursday ? I do not see the usefulness of decreasing the number of measurements by a factor of two.

p.14 l.21-24 The uncertainty analysis of the fraction given in this part of the paper is

missing, e.g. the sensitivity to a different threshold in the ozone increase (20% instead of 10%) or to a missing criteria (trajectories or low humidity)

p.16 l.15. It is not true to say that ozonesonde yield a substantial lower temporal coverage. For a ten year period, weekly launches (several stations in Europe are doing more than 1 launch per week) already provide as many independent profiles (520 profiles) as the IFU ozone lidar from 2007 to 2016 (585 measurement days). In fact I think the same analysis could be made for the european ozonesonde stations. Indeed it is recognized in the discussion p.17 l.13.

p.17 l.5 In fact direct intrusion is even decreasing at low altitude according to Fig. 1.

p.17 l.16-24 Other mechanisms must be included in the analysis like lower frequency of fast vertical transport of low level boundary layer ozone and longer lifetime of free tropospheric layers (see Colette et al. GRL 2006)

p.17 l.25-33 I do not undertand the meaning of this paragraph. It sounds like if the authors doubt about the methodology based on the ozone/RH joint analysis which is the backbone of this work.

Table 1: Specify that this table is for the number of O3 lidar data.

Fig. 3 Dark blue dots (start points in stratosphere are difficult to read). Black dots are for September 29 while the actual lidar data are shown on October 1st in Fig. 2. Why not providing the map for t0+6days ?

Fig.11 Caption says trajectories are calculated on June 28th while ozone lidar data are on May 28th, please clarify.

---

## Referee Comment (RC2) · Anonymous Referee #2 · 30 Sep 2019

**Review of "Very high stratospheric influence observed in the free troposphere over the Northern Alps – just a local phenomenon?" by Trickl et al.**

**General Comments:**

This paper presents the influence of stratospheric intrusions of high $O_3$ and low water vapor at/near Zugspitze summit station with surface, and $O_3$ and water vapor lidar data. The authors give several examples of the types and transport pathways of the intrusions, and summarize the frequency of the intrusions in ten years of data at Zugspitze. They find that 84% of days analyzed contained evidence of a stratospheric intrusion, which surprised both me and the authors. The pronounced summer maximum of "strong" intrusions (Fig. 14) was also counterintuitive. I think the authors will need to provide more evidence to support these extraordinary numbers, which seem to be at odds with many other studies, regardless of criteria used to identify intrusions (there is discussion of this in Section 4).

In particular, I strongly encourage the authors to at least perform a cursory intrusion analysis with the nearby Hohenpeissenberg ozonesonde data set (~3 profiles per week). That site is approximately 50 km north of Zugspitze, so the results should be very similar. Also, there should be more discussion on the peculiar summer maximum of "strong" intrusions. Is it possible to provide average lidar $O_3$ profiles on intrusion days, with the $O_3$ profile centered on the $O_3$ maximum (e.g. a profile in $O_3$-maximum relative altitude coordinates)?

I enjoyed the presentation of Section 3.2, but at least one of the sections is unnecessary and mostly irrelevant to the current $O_3$ study (#6: Volcanic Influence for example).

Does the title of the paper refer to the fact that the authors find much more frequent stratospheric influence over Zugspitze compared to studies in other locations? Please clarify.

Overall however, the paper is written well, enjoyable to read, and there are few technical mistakes.

**Recommendation:**

The paper itself requires mostly minor edits, but the authors first need to provide examples supporting the extraordinary frequency of stratospheric influence at Zugspitze cited here. This will take a more "major" effort.

**Technical/Line-by-line Comments:**

Page 1, Line 32: Please cite the TOAR paper on $O_3$ trends by Gaudel et al. (2018).

Page 1, Line 37: As you mention later in this paper on Page 3, the Montsouris measurements are in serious doubt (Tarasick et al., 2019; TOAR "Observations" paper). They should not be used to estimate past influence nor changes to the stratospheric influence on tropospheric $O_3$ amounts.

Page 2, Line 34: What does "mostly of the northern hemisphere" mean? Please clarify and rewrite.

Page 2, Line 37: "does by far not match" does not make sense. Please rewrite.

Page 3, Line 11: Delete "this"

Page 3, Line 21: Change "clear" to "distinct"

Page 3, Line 30-35: This discussion could be rewritten without citing the Montsouris data.

Page 4, Line 8: Change "the privilege of" to "limited to"

Page 4, Line 9: Change "were" to "was"

Page 4, Line 11: Please rewrite "Below 50 N the stratospheric layers were limited to this range." What does this refer to?

Section 2.1.3: Please provide a table summarizing the data sets used here including site, lat, lon, elevation, and period of data used in this study.

Section 2.2: A table could also clarify the multiple trajectory models and for what situations they are used here. It is currently confusing when and why each trajectory model is applied.

Section 2.2.2: For transport times >4 days, why not use LAGRANTO with archived ERA-Interim output? Again, the table would help clear up these questions.

Page 8, Line 15: Is this "reanalysis" data GDAS? What resolution (.5 or 1 deg?)

Page 8, Line 20: Please rewrite this sentence.

Page 9, Line 1: Please use "days" or some suitable substitution instead of "data files"

Figure 2: Why is the $O_3$ in the black profile so low above 12 km?

Figure 3: I find this very hard to interpret and have no idea what is going on in this figure. Is there a better way to present clusters of trajectories indicating where the air over Zugspitze at the time of the $O_3$ profiles originated?

Page 11, Line 8: Use of "dominates" in this title requires statistics for support.

Figure 7: Can you match the vertical scales of these plots so the altitudes are aligned?

Page 12, Line 1: Please reword this title i.e. "has been mostly high"

Page 12, Line 3: As mentioned in the General Comments, is it possible to calculate an $O_3$-maximum centered average $O_3$ profile to indicate the general conditions on these days?

Page 12, Line 9 (#6): This section seems out of place and could probably be removed.

Page 12, Line 27: I do not follow this logic. How have you shown that $O_3$ layers are more likely to descend toward the ground at night?

Page 13, Line 1: Please rewrite "just eight-day." What is FLEXTRA? Another trajectory model?

Page 13, Lines 1-2: Delete "during already"

Page 13 and Figs 11 and 12: Running HYSPLIT ensemble back trajectories for a single altitude over Zugspitze would be more convincing than single trajectories over this very long timescale. The same can be said for other HYSPLIT examples used in this paper. Also, a model should not be used to indicate high $O_3$ values over the US. In fact, the actual observations show that $O_3$ was not particularly high near the surface (www.airnowtech.org data from 0z 24 May 2015):

[Figure]

The Huntsville, AL, ozonesonde from 23 May 2015 at 18z also shows that $O_3$ in the boundary layer was generally ~60 ppbv. Please rethink the supporting arguments for this example.

Page 14, Line 11: Change "similarly" to "similar"

Page 14, Line 13-15: I'm not sure what this sentence means or is trying to argue.

Page 14, Line 22: This factor of 20 difference between two studies is part of what needs to be explained more as mentioned in the General Comments.

Page 15, Line 11: This is the most surprising result of the paper. I did not expect a summer maximum, and I certainly did not expect the intrusions to be mainly of the "strong" (as you define it) variety.

---

## Author Comment (AC1) · 18 Nov 2019

Reply to the reviewers´ reports on

**Very high stratospheric influence observed in the free troposphere over the Northern Alps – just a local phenomenon?**

By T. Trickl, H. Vogelmann, L. Ries, M. Sprenger

ACP-2019-588

Thomas Trickl, November 18, 2019

We thank the reviewers for very carefully reading the manuscript which has been helpful indeed. I have tried to follow the recommendations as far as possible. The view of Secs. 3.3 and 3.3 are conflicting in both reports. Here, we decided to stay with our own version, with some modifications. We think that describing typical situations is an important part of an observational paper.

This paper is not a final answer to the open issues of STT. It emerged from a routine analysis of ozone data that revealed an astonishingly high frequency of elevated-ozone layers. This stimulated a closer look at the sources of these layers that intensified with over the years. It is time to publish these interesting findings. There is no question that more work is needed in the future that arises from the questions that resulted from the present study.

In the following the paragraphs of the reports are written in Italics, the replies normal.

***Rewiew 1:***

*The paper deals with the analysis of a 9-year lidar data base (ozone UV DIAL and water vapour) to discuss the statistical occurrence of stratospheric intrusion above Southern Germany. The objectives are to extend previous estimate of the fraction of intrusion days derived from mountain top observation at 2962 m. Indeed the lidar data base appears well suited for such a task. Interesting results are discussed in section 3.4 and Fig. 13 and 14, and they deserved to be published. However the paper is not very well written with many digressions not necessary to discuss the results of the statistical analysis. The section 3.1 which is critical to understand the methodology, is not very clear and a table with the list of criteria to identify stratospheric intrusions must be provided. It is hard to see if the water vapour lidar is actually used for the relative humidity assessment. Section 3.2 and 3.3 are either very qualitative or out of scope and could be easily shortened or removed. Overall the paper must be published but with a major revision of the presentation. There are also many self citation to previous works which are not always necessary. Finally it is a pity a similar approach is not applied to the Hohenpeissenberg ozonesonde data base to alleviate error due to ozone/humidity layer mismatch and to obtain a complementary statistical study of intrusions in the same region.*

*Specific comments*

*p1 l37: Tarasick et al. 2019 (accepted for publication in Elementa) recently show that the 10-ppb figure from early ozone observations is strongly underestimated by a factor of 2. This will increase the estimate of transport from the stratosphere when using such a number. Please acknowledge this likely underestimate in the paper*

As a co-author of Tarasick et al. I have been aware of this. In fact, the higher values discussed in that paper are mentioned on p. 3, line 33-34, where also the paper is cited. It makes more sense to place it here since we proceed from older findings to more recent ones.

*p2 l10: add Ebel et al. for regional model studies https://doi.org/10.1016/S1352-2310(97)00063-0*

The link given leads me to Elbern et al. (1997) which is already cited on P. 2 (line 2). Their approach is based on data. Here, model results are cited. Ebel et al., indeed, made several model assessments of tropopause folds and fold statistics. However, I did not find a paper on the full budget from this group.

*p2 l29¢ a: Colette et al. 2006 https://doi.org/10.1029/2006GL025793 also analyzed the lifetime and mixing of laminae in the free troposphere. Add citation*

This is an interesting paper! However, it is based on FLEXPART calculations with model assumptions for mixing and, as far as I see, no indication of finding an overestimate of mixing. Trickl et al. (2014) give observational evidence for very low mixing based on 80 days with deep STT that the mixing schemes are most likely exaggerating. There are also case studies elsewhere confirming this (cited in our 2014 paper), but there is no study over that many cases. The 2016 paper is an extension of the 2014 study.

I tried hard to find a way to include (Colette et al., 2006) in an acceptable way. At the end I decided to give up.

*p3 l5: add Kowol et al. 2000 https://doi.org/10.1029/2000GL011369d where ozone exchange near the STJ is quantified.*

Thank you for this reference! However, I could not find any hint on the subtropical jet in this paper. Instead, I found in my files a GRL paper by Kowol-Santen and Ancellet (Mesoscale analysis of transport across the subtropical tropopause, 2000) that I add to the STJ paragraph. The paragraph deals about the importance of STT in the subtropics and I just cited papers revealing strong events. In order to include that paper I had to add a statement also some more observational publications regarding the STJ source latitudes.

*P3. l32: O3 precursor did not decline everywhere. Ozone still increases in Eastern Asia (see P arrish et al. ACP 2012)*

Thank you! This is quite obvious: I added "over Europe".

*p3 .l22: Better to show a figure with the Zugspitze data up to 2013 for a better comparison with the lidar time period 2007-2016. Considering the rise of tropospheric ozone in Fig. 1 (30→46ppb), the direct ozone intrusion is actually decreasing from 1978 to 2004 and only the indirect contribution is increasing form 1/3 to 1/2 .*

This is a reasonable suggestion. Unfortunately, the re-analysis of the Zugspitze ozone is still not done! This considerable effort is now planned for next year. It is very likely that I am going to include also ozone data from nearby UFS (2670 m) until present. I cannot provide the trend for the direct component since I do not possess the numbers: Dr. Scheel, who prepared the figure, passed away in 2013. He claimed a slight positive trend, and, from the figure, cannot see the opposite.

*P4, L18-22: After a nice overview of stratospheric ozone transport at different latitude and altitude, one would expect a better description of the actual goals of this work in the introduction and how the authors will fill the gaps identified in the introduction.*

I added a brief introduction to the individual sections at the end.

*p.6 l13-20 Only half of the lidar record corresponds to the in-situ data set. How do the authors deal with this ?*

There is no question that more lidar measurements would be desirable. The limitation to about 500 ozone measurements per year in years without technical problems is caused by the time still needed for the data evaluation.

The in-situ data set consists of half-hour averages listed with very few gaps. Thus, the data coverage in our 2010 five-year assessment of STT is much higher. This is discussed in the chapter on statistics.

*p.6 l33 to 37 Do you mean that RH from radiosonde are used in conjunction with the O3 lidar to identify intrusion ? What is the uncertainty due to layer mismatch or radiosonde poor sensitivity in very dry layers ? Why are the Hohenpeissenberg ozonesonde not used in the analysis of your case studies ?*

RH from sondes has, indeed, been routinely used. The overlap of the ozone and the water-vapour DIAL is limited (120 common measurement days, which is now mentioned). Thus, the sonde data were highly necessary. In most cases the sonde RH verifies at least one intrusion measured at Garmisch-Partenkirchen, just slightly shifted in altitude. I extended the text in this subsection for more clearness.

As mentioned, the sensitivity and reliability of the RS 92 sonde is very high. Unfortunately, the German Weather Service replaced this sonde type by the RS 41 sonde right after the range of years discussed here that has a wet bias of several per cent RH in dry layers.

The ozone-sonde ascents at Hohenpeißenberg are limited to Monday, Wednesday and Friday during the warmer half of the year, and to Monday and Friday during the other one, and to just 6 a.m. on all these days. They have not been used for the routine lidar sounding here (RH), just in very few cases for ozone comparisons, requiring personal request.

The use of Hohenpeißenberg ozone soundings is an interesting option for future work, as proposed in the Discussion section. This should be done by the colleagues at Hohenpeißenberg who created the data, or in co-operation.

*p.7 l25-30 Is a paragraph advertising for availability of the "intrusion hit tables" really necessary in this paper ? Better to discuss here Fig. 3 which is a good example of a LAGRANTO output.*

In critical cases I have inspected the hit tables and I do not understand why I should not mention them here. A discussion of the LAGRANTO results clearly belongs to the "Results" section. This is why I have chosen Fig. 3 here since this is just one of two cases presented in that section where the descent occurred in four days as required for the operational forecasts. The other case would be that of Fig. 4, but I did want to blow up that subsection by an analysis.

I modified the description of Fig. 3.

*P8. l15-18 The author cannot claim the re-analysis are better than operational analysis without a thorough discussion about the methodology to compare the two meteorological fields. The following sentence about validation of backtrajectories with observations of ozone is good enough to justify the use of HYSPLIT in this paper. Is the diabatic air mass vertical motion in the UTLS properly accounted for when using the analysed vertical wind in 15-day trajectory calculation ?*

I do not claim anything on the models! I just mention that we prefer to use the "reanalysis" data because of practical experience over many years. E.g., in our 2015 paper in ACP where fire positions and CALIPSO images were used to verify the correctness of the re-analysis mode. We discuss one example (Fig. 11) also in this paper. I added one sentence to refer to this example.

For an earlier publication (2015) I asked the HYSPLIT team about the model type: "Model Vertical Velocity" means full three-dimensional.

*p.9 This section is quite important because it is the backbone of the statistical analysis. It deserves a summary table to list the different criteria to identify a stratospheric intrusion: ozone and RH thresholds, from which instruments, need for forward of backward trajectories. The discussion about the humidity values within stratospheric intrusion is hard to follow. I understand a 10% RH threshold is taken. But the following discussion focuses on a list of case studies with smaller RH values. Please rewrite the paragraph L24 to 34 (emphasize that 10 % is good enough and avoid multiple references to your past studies). It is not clear if the aerosol lidar data are actually used and how. If not discussion about the aerosol lidar in section 2.1 is unnecessary and the lengthy presentation of Saharan dust or volcanic/fire aerosol transport events is not needed in section 3.2.*

Thank you very much for this comment! It is strange that I did not realize that the criteria were not explained clearly. I made several changes in that section that also removed some of the citations.

I do not see anything about the aerosol measurements on P. 9. Aerosol has not been used in the STT analysis. Aerosol measurements are just mentioned in Sec. 3.2 to underline two types of observations that are not much treated in the literature. The coexistence of Saharan dust and intrusions occurred on a total of 67 days. Thus, the advection via North Africa is an important pathway worth mentioning. This part is roughly half a page and not excessive. I slightly shortened it. The downward transport of stratospheric aerosol in intrusions is a key depletion mechanism. For us, this is an important field because of our measurements of stratospheric aerosol since 1976.

*p.9 l.1 Better to emphasize the number of ozone measurements days (585) which is a better proxy to assess how meaningful is the statistical study. For example in Fig. 10 on May 28 there are only two clearly independent profiles while 7 are reported. Give also the number of water vapour lidar observations coincident with the ozone data.*

I added the number of measurement days here as well as those of the water-vapour DIAL farther below on that page.

*p.9.L13 What do you mean by ozone as a secondary indicator ?*

This sentence is confusing and was removed. This statement was important just for the analysis published in 2010.

*P9.L15 A 10% ozone increase seems a very low threshold considering first the measurement accuracy (5%-10%) and second the ozone variability due other transport processes (horizontal differential advection, convection).*

10 % means about 5 ppb which can be distinguished in the lower troposphere. In winter, the absorption by ozone (40 ppb) is low and the data noise diminishes also at higher altitudes. In the upper troposphere the ozone rise in intrusion layers is typically higher anyway.

A rise by just 5 ppb is rare indeed. I first had a hard time to accept these cases!

I modified this part.

*Section 3.2 and 3.3 These two sections are way too long and is of little value for the analysis developed in section 3.4. I recommend to keep only quantitative information needed for the statistical study e.g. two examples of ozone vertical profiles: a thin and thick intrusion and the ancillary information to identify the stratospheric origin. Quantitative information of some features discussion section 3.2 can be indeed developed: - very intense intrusion are rare (define and give occurrence fraction) - thin layers are frequent (same recomm.) - slow descent are frequent above 5 km (same recomm.) - intrusion does not penetrates in PBL (how many cases ?) Discussions about the dust cases or the volcanic episode are out of scope and do not add a new perspective in the analysis. I get the feeling the authors wish to emphasize the lidar capabilities of monitoring both aerosol and ozone without making an actual use of this capability for a better assessment of the stratospheric intrusion frequency in Southern Germany. P.14*

I understand that Reviewer 1 is primarily interested in the statistical analysis. However, this is an observational paper on STT, and the most important observational findings must be described, which was strongly confirmed by my co-authors. A reader not interested in this is free to omit this part. Section 3.2 covers less than three pages which is not excessive at all.

The topic of intercontinental transport (Sec. 3.3) is an important issue when analysing high-ozone layers in summer and cannot be treated in a few lines. During the first years of our studies of intercontinental transport (Stohl and Trickl, 1999; Trickl et al., 2003) there was always the question about additional stratospheric contributions in the relevant layers. With growing length of the Lagrangian calculations this was more and more verified.

The "superfluous" parts are not written for emphasizing the lidar capabilities, they aim at emphasizing what can be learnt from combining different methods. The value of the dust observations and the stratospheric aerosols is pointed out above. I hope that the data can be analysed also under different aspects in the future.

I made several changes.

*p. 14 l.5-10 Do you mean that the water vapour lidar is not used ? Please clarify how you truly derive the RH needed for the stratospheric intrusion analysis. This is a critical parameter and large bias can occur if the two measurements are not co-localized.*

We did compare the data from DIAL systems. I am wondering how the corresponding statement got lost. I added this information on P. 9. It is interesting to note that RH = 10 % seemed to be a natural threshold in the data which did not require to drop many data.

Here, we are discussing the possibility of quantifying the ozone transfer. This does only work if side-by-side water-vapour measurements are available. The widths of the dry layers could then determine the range where ozone can be identified as stratospheric. Unfortunately, the DIAL was not always operated on the same day as the ozone DIAL.

I modified the text.

*p.14 l16-20 Please explain what you are doing here. Why do you focus on data taken on monday and thursday ? I do not see the usefulness of decreasing the number of measurements by a factor of two.*

This was done to verify that there is no bias in the results, e.g., by selecting the measurement according to STT forecasts (this was, by the way, not the case!). Monday and Thursday were chosen for this purpose within EARLINET and, as an EARLINET station, we adopted this schedule. Of course, as mentioned in the paper, this deteriorates the quality.

I changed the sequence in this paragraph.

*p.14 l.21-24 The uncertainty analysis of the fraction given in this part of the paper is missing, e.g. the sensitivity to a different threshold in the ozone increase (20% instead of 10%) or to a missing*

Do you mean "sensitivity analysis"? As pointed out the ozone threshold is seen as a minor criterion. The number of cases with a 10 % exceedance of the background are really very small. A slightly higher threshold would, therefore, just slightly reduce the fraction in winter. I added a sentence regarding this in Sec. 3.1. Figure 14 shows that a threshold of 15 ppb is already met in 30 % of the winter cases. An increase of ozone in a dry layer is necessary to make sure that we are not probing air from just below the tropopause.

*p.16 l.15. It is not true to say that ozonesonde yield a substantial lower temporal coverage. For a ten year period, weekly launches (several stations in Europe are doing more than 1 launch per week) already provide as many independent profiles (520 profiles) as the IFU ozone lidar from 2007 to 2016 (585 measurement days). In fact I think the same analysis could be made for the european ozonesonde stations. Indeed it is recognized in the discussion p.17 l.13.*

You are comparing profiles and days! There is a high intraday variability! I admit that the data coverage of our lidar measurements has not been ideal (this would be full 24 h of measurements during fair-weather) because of time limitations for the data evaluation. But this can be improved!

Nevertheless, replaced that sentence by "Routine operations are typically limited to just a single launch on one to three days per week". At Hohenpeißenberg launches take place on three days during the warm season and on two days during the rest of year.

*p.17 l.5 In fact direct intrusion is even decreasing at low altitude according to Fig. 1.*

As mentioned above, I do not confirm this.

*p.17 l.16-24 Other mechanisms must be included in the analysis like lower frequency of fast vertical transport of low level boundary layer ozone and longer lifetime of free tropospheric layers (see Colette et al. GRL 2006)*

I agree. This is why I included Sec. 3.3. All is very complex. Air streams from the stratosphere and the PBL are merging on their way around the northern hemisphere (see Cooper et al.). This opens possibilities for future activities. Here, we just focus on the cases with low RH, i.e., with low contributions from remote PBLs.

I do not know how to add more possibilities to that paragraph without losing clearness. I decided to add "to some extent" to the explanation discussed.

*p.17 l.25-33 I do not undertand the meaning of this paragraph. It sounds like if the authors doubt about the methodology based on the ozone/RH joint analysis which is the backbone of this work.*

Here, we are discussing concentrations and not fractions! This paragraph just repeats an earlier one.

*Table 1: Specify that this table is for the number of O3 lidar data.*

Added!

*Fig. 3 Dark blue dots (start points in stratosphere are difficult to read). Black dots are for September 29 while the actual lidar data are shown on October 1st in Fig. 2. Why not providing the map for t0+6days ?*

This figure will be printed over two columns as in previous papers. The black dots for the trajectories leading to our site are located over Russia. Thus, they arrive roughly a noon on September 30. The residual time shift is explained in the text: the shift was chosen for clearness.

*Fig.11 Caption says trajectories are calculated on June 28th while ozone lidar data are on May 28th, please clarify.*

Changed!

***Review 2:***

**Review of "Very high stratospheric influence observed in the free troposphere over the Northern Alps – just a local phenomenon?" by Trickl et al.**

***General Comments:***

*This paper presents the influence of stratospheric intrusions of high O3 and low water vapor at/near Zugspitze summit station with surface, and O3 and water vapor lidar data. The authors give several examples of the types and transport pathways of the intrusions, and summarize the frequency of the intrusions in ten years of data at Zugspitze. They find that 84% of days analyzed contained evidence of a stratospheric intrusion, which surprised both me and the authors. The pronounced summer maximum of "strong" intrusions (Fig. 14) was also counterintuitive. I think the authors will need to provide more evidence to support these extraordinary numbers, which seem to be at odds with many other studies, regardless of criteria used to identify intrusions (there is discussion of this in Section 4).*

There is no doubt about about our analysis which was done very carefully and took several hours per measurement day. I think we have discussed that a significant number of studies that are not too far away from our findings, at least during certain seasons.

*In particular, I strongly encourage the authors to at least perform a cursory intrusion analysis with the nearby Hohenpeissenberg ozonesonde data set (~3 profiles per week). That site is approximately 50 km north of Zugspitze, so the results should be very similar. Also, there should be*

*more discussion on the peculiar summer maximum of "strong" intrusions. Is it possible to provide average lidar O3 profiles on intrusion days, with the O3 profile centered on the O3 maximum (e.g. a profile in O3-maximum relative altitude coordinates)?*

The analysis of the Hohenpeißenberg data has been seen by us as very important since the balloon ascent take place independent from the weather conditions and since $H_2O$ and $O_3$ is measured on the same platform. We already emphasize on this important issue on p. 17, lines 13-15. Analysing the Hohenpeißenberg data would be the job of the colleagues from the German Weather Service who have generated them, or would at least require a co-operation. For us, this is a completely new project that is not funded at this time, myself being retired.

Providing average $O_3$ profiles would smooth away the information. As mentioned we have not been able to identify all the elevated-ozone layers that simultaneously pass over our site on a single summer day. This would require a more sophisticated approach based on FLEXPART, beyond our current capability. Also there is not enough information on dry layers since measurements of the $H_2O$ lidar do not fully coincide with those of the O3 DIAL, and the radiosonde stations are rather remote to hit all the dry layers passing over Garmisch-Partenkirchen. Thus, we have limited ourselves to the kind of statistics presented, just clearly identifying at least one STT layer on a given day. This is just the first step towards a more comprehensive analysis.

*I enjoyed the presentation of Section 3.2, but at least one of the sections is unnecessary and mostly irrelevant to the current O3 study (#6: Volcanic Influence for example).*

This topic may be irrelevant for the statistical analysis of STT. However, it highly relevant in the context of STT. We have seen aerosol in quite a number of intrusions after eruptions, and now also after the strong fires in British Columbia in 2017 (which is now also mentioned).

*Does the title of the paper refer to the fact that the authors find much more frequent stratospheric influence over Zugspitze compared to studies in other locations? Please clarify.*

The title end with a question mark! We discuss a number of publications that confirm that large fractions of STT exist also elsewhere, but, in the absence of a better data coverage, this is cannot be extrapolated to the entire globe. Jet streams do not exist everywhere. We think that existing Lagrangian model studies must be extended to many more days of descent. Different amounts of mixing must be included.

*Overall however, the paper is written well, enjoyable to read, and there are few technical mistakes.*

**Recommendation:**

*The paper itself requires mostly minor edits, but the authors first need to provide examples supporting the extraordinary frequency of stratospheric influence at Zugspitze cited here. This will take a more "major" effort.*

The rather quantitative analysis of intrusions hitting the Zugspitze summit was published by us in ACP in 2010. As mentioned, we do not see major discrepancies with that analysis in our new study. A full re-analysis of (like that leading to Fig. 1), based on the new criteria in our 2010 paper is planned for next year.

As to the free troposphere, I do not know what "examples" means. We have given several examples in Sec. 3 and in preceding publications.

**Technical/Line-by-line Comments:**

*Page 1, Line 32: Please cite the TOAR paper on O3 trends by Gaudel et al. (2018).*

Good idea, done! The first version of our paper was written before that paper was published and we forgot to include it later on.

*Page 1, Line 37: As you mention later in this paper on Page 3, the Montsouris measurements are in serious doubt (Tarasick et al., 2019; TOAR "Observations" paper). They should not be used to estimate past influence nor changes to the stratospheric influence on tropospheric O3 amounts.*

The introduction proceeds historically. Thus, it starts with the old assumptions. What we did is not wrong: A comparison with Montsouris does result in a negative anthropogenic ozone trend in the 1990s. Thus, there is some doubt left that is confirmed by the TOAR paper.

*Page 2, Line 34: What does "mostly of the northern hemisphere" mean? Please clarify and rewrite.*

I think it refers to the geographical coverage of MOZAIC. This phrase is not necessary and is now deleted.

*Page 2, Line 37: "does by far not match" does not make sense. Please rewrite.*

Changed!

*Page 3, Line 11: Delete "this"*

Deleted!

*Page 3, Line 21: Change "clear" to "distinct"*

Changed!

*Page 3, Line 30-35: This discussion could be rewritten without citing the Montsouris data.*

I like the way it was written. The introduction started with Montsouris which had been a landmark in the history of ozone research. Here, we see some indication that the Montsouris value mighty be too low, which is confirmed by the publications cited in the TOAR report.

*Page 4, Line 8: Change "the privilege of" to "limited to"*

Changed!

*Page 4, Line 9: Change "were" to "was"*

Changed!

*Page 4, Line 11: Please rewrite "Below 50 N the stratospheric layers were limited to this range." What does this refer to?*

Changed to "altitude range"!

*Section 2.1.3: Please provide a table summarizing the data sets used here including site, lat, lon, elevation, and period of data used in this study.*

I do not understand! Just data from Garmisch-Partenkirchen were taken as the primary basis of our work. The Zugspitze sites are just a few kilometres away from here as explicitly pointed out. Also the distance to the sonde stations is mentioned. The lidar data for ozone are specified in Table 1. They determine the field used for the analysis. Until the end of 2014, when the laser of the water-vapour lidar finished operation, the humidity data from this lidar were the primary basis. I added the number of simultaneous measurement days (120 between 2007 and 2014). Nevertheless, on all ozone sounding days exhibiting conspicuous structures sonde data were used. This does not justify an additional table!

We give an improved description of the data selection.

*Section 2.2: A table could also clarify the multiple trajectory models and for what situations they are used here. It is currently confusing when and why each trajectory model is applied.*

Trajectories were used on each measurement day with potential STT layers. If LAGRANTO trajectories (four days) explained a specific observation they were used. HYSPLIT trajectories have been calculated on all days exhibiting dry layers with elevated ozone, in addition to LAGRANTO.

This is really not difficult to understand and does not justify a table! Such a table would of excessive size anyway: I have stored thousands of trajectory plots.

I changed the text in several sections for more clearness.

*Section 2.2.2: For transport times >4 days, why not use LAGRANTO with archived ERA-Interim output? Again, the table would help clear up these questions.*

This would have been a hard extra job for our partner at ETH and practically unrealistic. HYSPLIT has been fully accessible. Thus, the analysis was possible in near real time.

*Page 8, Line 15: Is this "reanalysis" data GDAS? What resolution (.5 or 1 deg?)*

No! As mentioned in the section on model description (2.2.2), GDAS is a high-resolution model that becomes available with a delay of one day. "Reanalysis" meteorological data are evaluated with a delay of one month. Due to near-real-time archiving during the final three years the re-analysis mode of HYSPLIT was just used if the data analysis was delayed. Thus, as explicitly mentioned, re-analysis trajectories were just used (later on) if the GDAS trajectories failed to explain an intrusion. The number of these cases was low.

The better quality of the reanalysis trajectories was mentioned by us previously. The HYSPLIT web site specifies different resolutions for different parameters, all beyond 1.5 degrees. This has really puzzled me, but we had to accept this fact. I prefer not to cite this value explicitly since I am not sure about these details.

*Page 8, Line 20: Please rewrite this sentence.*

Rewritten in a less complicated manner!

*Page 9, Line 1: Please use "days" or some suitable substitution instead of "data files"*

Days would be too much! New: individual measurements

*Figure 2: Why is the O3 in the black profile so low above 12 km?*

I add: "The ozone minimum at 12.2 km (7:52 CET)) is close to the upper end of the operating range of the lidar for these extreme concentrations, but is clearly visible and ascribed to the tropopause. This is justified by the position of the Munich tropopause that descended from 13341 m to 11903 m from midnight until noon."

*Figure 3: I find this very hard to interpret and have no idea what is going on in this figure. Is there a better way to present clusters of trajectories indicating where the air over Zugspitze at the time of the O3 profiles originated?*

I agree: Several intrusions intersect which causes complexity. However, a close look clearly shows the main path of the trajectories relevant for the observations that I now describe in more detail. The pathway is exactly confirmed by HYSPLIT.

We need one example for LAGRANTO in this paper. Here, the descent occurs within just five days (or less) which makes this case most suitable.

*Page 11, Line 8: Use of "dominates" in this title requires statistics for support.*

I made a statistical analysis for the three years with best coverage by measurements. The variability is high, due to the changing weather patterns. The fraction of intrusion days with at least one Type-layer is 48 % ± 13 %. Some uncertainty remains since the exit point of a trajectory from the stratosphere is not always clearly discernible. East of 80º W over Canada the air mass would be of Type 5. I prefer to add the result of this analysis to the chapter on statistics (and refer to this section), this makes more sense.

*Figure 7: Can you match the vertical scales of these plots so the altitudes are aligned?*

Done! Indeed, a 250-m vertical displacement between both lidars can be better seen in this way that is typical and ascribed to orographic lifting above the mountain.

*Page 12, Line 1: Please reword this title i.e. "has been mostly high"*

I do not understand! I replaced "mostly" by "frequently".

*Page 12, Line 3: As mentioned in the General Comments, is it possible to calculate an O3-maximum centered average O3 profile to indicate the general conditions on these days?*

As mentioned: These examples are really typical. Averaging would not change this. What is needed is a better analysis of the details of the high-ozone part of the profiles. This would change the scope of this paper.

*Page 12, Line 9 (#6): This section seems out of place and could probably be removed.*

I do not agree: This is an important topic of downward transport from the stratosphere.

*Page 12, Line 27: I do not follow this logic. How have you shown that O3 layers are more likely to descend toward the ground at night?*

As mentioned there is a strong indication of this, descent of ozone peaks to low altitudes have been observed. It would be interesting to study this in detail in the future. For this we need $H_2O$ profiles in the PBL that may become available with a recently completed lidar.

I wrote a new introduction to this paragraph pointing out the relevance of this topic.

*Page 13, Line 1: Please rewrite "just eight-day." What is FLEXTRA? Another trajectory model?*

Changed to make "just" clearer! As indicated, FLEXTRA is another trajectory model. I add a citation.

*Page 13, Lines 1-2: Delete "during already"*

I deleted "already".

*Page 13 and Figs 11 and 12: Running HYSPLIT ensemble back trajectories for a single altitude over Zugspitze would be more convincing than single trajectories over this very long timescale. The same can be said for other HYSPLIT examples used in this paper. Also, a model should not be used to indicate high O3 values over the US. In fact, the actual observations show that O3 was not particularly high near the surface (www.airnowtech.org data from 0z 24 May 2015):*

*The Huntsville, AL, ozonesonde from 23 May 2015 at 18z also shows that O3 in the boundary layer was generally ~60 ppbv. Please rethink the supporting arguments for this example.*

As said in the chapter introducing the model, ensemble trajectories would be desirable, but were by far not long enough for our study. Thus, I spent a lot of time in calculating numerous trajectory bundles and picked just the most representative ones for the figures shown.

I appreciate this kind of help. I asked American colleagues for providing ozone values and received the ozone map shown. Of course, observational data would be more convincing than model data. Huntsville is, indeed, in southern part of the "hot" zone modelled. 18 z is in the morning, but I do not know I examined ozone maps also for Kentucky and Illinois and found an ozone rise exactly on May 23, covering the entire states, whereas in Alabama there were just local maxima. But the EPA values are not quantitative ("elevated" could mean roughly 70 ppb). We do not know if further ozone formation took place during the rise of the air mass.

I add a sentence on this ozone rise on May 23 and replaced "highly satisfactory" by "rather satisfactory".

*Page 14, Line 11: Change "similarly" to "similar"*

Changed!

*Page 14, Line 13-15: I'm not sure what this sentence means or is trying to argue.*

I changed "variability" to "variation". The following sentence is changed for more clearness.

*Page 14, Line 22: This factor of 20 difference between two studies is part of what needs to be explained more as mentioned in the General Comments.*

A full quantification of this enormous factor would require a full re-analysis of the data used by Beekmann et al. This would be interesting, but is beyond the scope of this paper. The focus here is on our own data. In the "Discussion" section we try to give some qualitative explanations, and we suggest some re-analysis of long sonde series.

I made a few adjustments.

*Page 15, Line 11: This is the most surprising result of the paper. I did not expect a summer maximum, and I certainly did not expect the intrusions to be mainly of the "strong" (as you define it) variety.*

The summer maximum is not very pronounced. At least, in agreement with Beekmann et al., there is no minimum as in the lower troposphere. To fill that minimum, the assumption of an upper-tropospheric maximum is needed. The question is: Is this related to more pronounced dynamics in the tropopause region in summer? Why not: wider layers extracted from the lowermost stratosphere lead to higher ozone concentrations in the intrusions.

[revised manuscript text omitted]